# Predictably manipulating photoreceptor light responses to reveal their role in downstream visual responses

Qiang Chen, Norianne T Ingram, Jacob Baudin, Juan M Angueyra†, Raunak Sinha‡, Fred Rieke*

Department of Physiology and Biophysics, University of Washington, Seattle, United States

*For correspondence:
rieke@uw.edu

Present address: †Department of Biology, University of Maryland, College Park, United States; ‡Department of Neuroscience, University of Wisconsin-Madison, Madison, United States

## eLife assessment

This paper provides an **important** method that uses a computational model to predict photoreceptor currents in mammalian photoreceptors. By inverting the model, visual stimuli can be constructed to produce desired photoreceptor current responses. The authors provide **compelling** evidence that this approach can disentangle the effects of photoreceptor nonlinearities including light adaptation from downstream nonlinear processing, thus facilitating future studies of the higher visual system.

**Abstract** Computation in neural circuits relies on the judicious use of nonlinear circuit components. In many cases, multiple nonlinear components work collectively to control circuit outputs. Separating the contributions of these different components is difficult, and this limits our understanding of the mechanistic basis of many important computations. Here, we introduce a tool that permits the design of light stimuli that predictably alter rod and cone phototransduction currents – including stimuli that compensate for nonlinear properties such as light adaptation. This tool, based on well-established models for the rod and cone phototransduction cascade, permits the separation of nonlinearities in phototransduction from those in downstream circuits. This will allow, for example, direct tests of how adaptation in rod and cone phototransduction affects downstream visual signals and perception.

## Introduction

A central goal in neuroscience is to understand how behaviorally significant computations are implemented by cellular and synaptic mechanisms (see *Churchland and Abbott, 2016*). Such computations typically rely on a diverse collection of interacting circuit mechanisms, many of which are nonlinear (*Gollisch and Meister, 2010*; *Isaacson and Scanziani, 2011*; *Jadzinsky and Baccus, 2013*; *Zador, 2000*). Circuit outputs – which are often most amenable to experimental measurement – typically do not uniquely identify the role of specific circuit mechanisms. Hence progress toward understanding the mechanistic basis of circuit function requires better tools to manipulate specific mechanisms and identify their contribution to circuit output.

Sensory systems provide a clear example of these issues. Sensory processing is strongly shaped by the properties of the sensory receptors themselves and by post-receptor circuit mechanisms. Yet separating the contributions of receptor and post-receptor mechanisms to circuit outputs can be difficult, and computational models often make untested assumptions about their relative importance. For example, models for retinal processing generally assume that photoreceptor responses are linear

or near linear, despite decades of direct evidence that this is not the case (reviewed by *Burns and Baylor, 2001*; *Schwartz, 2021*). Adaptation to changes in mean light intensity provides a specific and important example. Both photoreceptors and post-photoreceptor circuits adjust response properties such as gain and kinetics to match the prevailing light inputs (reviewed by *Demb, 2008*; *Dunn and Rieke, 2006*), yet models for retinal outputs typically start by passing light inputs through a linear spatio-temporal filter; this architecture includes linear–nonlinear (*Chichilnisky, 2001*), generalized-linear (*Pillow et al., 2008*), and computational neural network (CNN) (*McIntosh et al., 2016*; *Turner et al., 2019*) models. Models with an initial linear filtering stage cannot capture the spatially local adaptation produced by photoreceptors. Tools that separate receptor and post-receptor contributions to adaptation are needed to better understand how this salient aspect of retinal processing works.

Here, we show how we can predictably manipulate the responses of rod and cone photoreceptors to causally probe their role in shaping responses of downstream visual neurons. We build on decades of work identifying and testing biophysical models for rod and cone phototransduction (*Nikonov et al., 2000*; *Pugh and Lamb, 1993*; *Rieke and Baylor, 1998a*; *Younger et al., 1996*). We show that existing phototransduction models, with appropriate parameters, can account for responses of rod and cone photoreceptors from both primate and mouse to a broad range of inputs. The resulting forward models can be used as a front end for encoding models for responses of downstream visual neurons, resulting in a better accounting for early time-dependent nonlinearities. We then show that these models can be mathematically inverted – enabling the design of stimuli that will elicit photoreceptor responses with specific desired properties such as a linear dependence on light input. Direct recordings of responses to these stimuli show that they work as designed. This approach provides a tool that can be used to predictably shape the photoreceptor responses and hence to casually dissect the impact of specific properties of the phototransduction currents on downstream visual signaling and perception.

## Results

### Biochemical model

We chose a model architecture based on the biochemical interactions that comprise the phototransduction cascade. We chose this model rather than empirical models (e.g. *Clark et al., 2013*) because it was more clearly connected to the mechanistic operation of the phototransduction cascade and because it was exactly invertible (see *Angueyra et al., 2022* for comparison of the biochemical model with empirical models).

Combined work in biochemistry, molecular biology, and physiology has produced a clear understanding of the operation of the phototransduction cascade (*Figure 1A*; reviewed by *Burns and Baylor, 2001*; *Rieke and Baylor, 1998b*; *Schwartz, 2021*). In brief, the phototransduction current flows through cGMP-gated channels in the photoreceptor outer segment. Levels of cGMP, and hence the phototransduction current, are maximal in darkness. Light activates G-protein-coupled receptors, which activate G-proteins and a cGMP-phosphodiesterase (PDE). The resulting decrease in cGMP concentration allows some cGMP-gated channels to close and decreases the current. Levels of cGMP are restored by synthesis of cGMP by a guanylate cyclase; cyclase activity is sped by a calcium-feedback pathway that strongly shapes light responses (*Burns et al., 2002*).

The biochemical understanding summarized in *Figure 1A* has led to several quantitative models for phototransduction (*Nikonov et al., 2000*; *Pugh and Lamb, 1993*; *Rieke and Baylor, 1998a*; *Younger et al., 1996*). We focus on a set of differential equations that form the core of these models. Our goal was to develop a model that generalized across stimuli rather than one in which the parameters provide accurate estimates of how specific components of the cascade work. Below we show that the model illustrated in *Figure 1A*, with appropriate parameters, captures the responses of rod and cone photoreceptors to a variety of stimuli. Most importantly for our purposes here, the models are sufficiently accurate to enable us to accurately predict stimuli that elicit desired photoreceptor responses, as tested in Figures 6–9.

Our phototransduction model has a total of 11 parameters (*Figure 1A*). Two of these ($q$ and $S_{max}$) can be expressed in terms of others using steady state conditions – for example the constraint that $dC(t)/dt = 0$ in steady state. Several others were fixed based on previous measurements and their

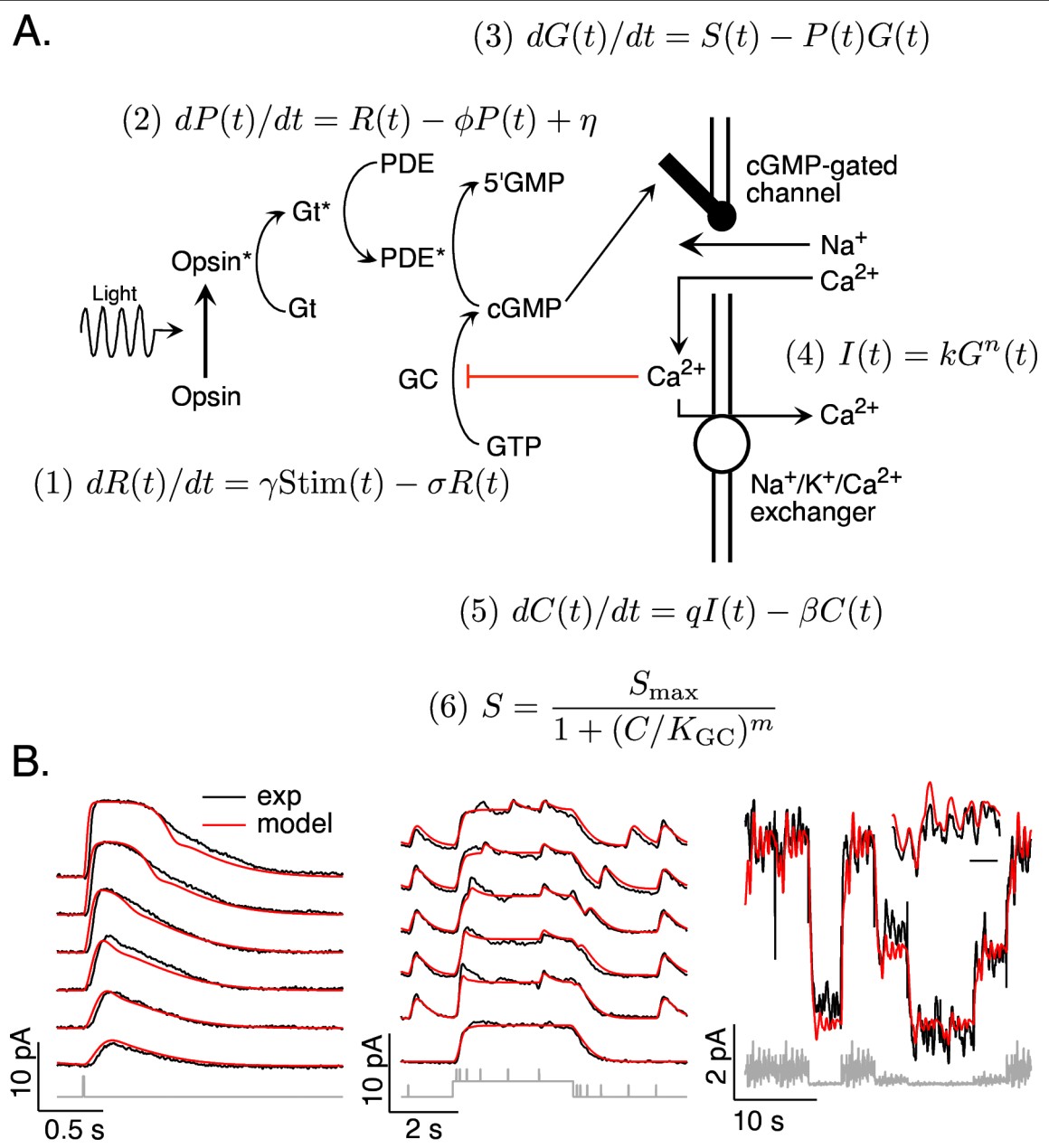

**Figure 1.** Model and fitting procedure. (**A**) Phototransduction cascade and differential equations that describe the operation of key components. Abbreviations are as follows: Stim: stimulus; R: receptor (photopigment) activity; P: phosphodiesterase activity; G: cGMP concentration; S: rate of cGMP synthesis; I: membrane current; C: calcium concentration. (**B**) Fits (red) of a model with four free parameters ($\gamma$, $\sigma$, $\eta$, $K_{GC}$) to the responses of a mouse rod (see *Table 1*). All measured responses are from the same rod, and all model responses use the same values for the free parameters. Responses to different flash strengths or flash timing have been displaced vertically in the left two panels. The weakest flash in the far left panel produced on average ~3 activated rhodopsin molecules, and each successive flash was twice as bright. The bottom trace in the middle panel was to a light step alone, while the other traces in this panel each had five superimposed flashes. The horizontal scale bar for the inset in B is 1 s.

relatively minor impact on model predictions; this included *k*, *n*, and *m* for both rods and cones (***Koutalos et al., 1995b***; ***Rieke and Baylor, 1996***) and $\beta$ for rods (***Field and Rieke, 2002***). The dark cGMP concentration was estimated from the measured dark current for each recorded cell. The receptor and PDE decay rates ($\sigma$ and $\Phi$) had near identical effects on predictions (see ***Pugh and Lamb, 1993***) and were constrained to be equal. These considerations allowed us to reduce the number of free model parameters to 4 for rods and 5 for cones. This choice reflects a balance of minimizing the

number of free model parameters while providing sufficient flexibility to capture key aspects of the light response.

The free parameters consisted of the gain $\gamma$ with which photons are converted to photopigment activity, the photopigment decay rate $\sigma$, the spontaneous PDE activation rate $\eta$, the constant $K_{GC}$ specifying the calcium sensitivity of the cGMP synthesis rate, and (for cones) the rate $\beta$ of $Ca^{2+}$ extrusion. This choice of free parameters is not unique given the similar impact of some parameters on model output (such as $\sigma$ and $\Phi$). But, as indicated below, it provided sufficient flexibility to capture measured responses to a variety of stimuli. We numerically identified optimal values for these parameters separately for each photoreceptor type (see Methods).

We were able to collect responses to a larger set of stimuli from individual rods than cones due to differences in recording techniques. We recorded from rod photoreceptors using suction electrodes, which allow for stable, long-lasting recordings. This allowed us to measure responses to a comprehensive set of stimuli from each individual rod. We recorded from cone photoreceptors using whole-cell patch clamp recordings. Light responses are stable in these recordings for a much shorter time (due to internal dialysis), and consequently we recorded responses to a single stimulus type from each individual cone. Despite these differences in data collection, we identified values for the free parameters in both rod and cone models by numerically minimizing the mean-squared error (MSE) between model predictions and measured responses (see Methods for more details).

## Fitting rod phototransduction models

We measured responses of mouse and macaque rod photoreceptors to a set of stimuli consisting of a family of brief flashes of different strengths, flashes delivered at several times relative to the onset and offset of a light step, and a 'variable mean noise stimulus' consisting of Gaussian noise with periodic changes in mean intensity (*Figure 1B*; see Methods for experimental details). The variable mean noise stimulus approximates the large and frequent changes in input that are characteristic of natural vision (*Frazor and Geisler, 2006*). These large (up to 30-fold) changes in mean intensity strongly engage photoreceptor adaptation, and the superimposed Gaussian noise probes sensitivity to rapidly varying inputs.

We recorded responses to this collection of stimuli for each rod in our dataset. We fit mouse and macaque rods separately, in each case seeking a common model (i.e. with the same parameters)

**Table 1.** Parameters and best fit consensus values for biophysical phototransduction models for each cell type (see Methods for fitting details).

| Parameter | Symbol | Units | Type | Photoreceptor type | | | |
|---|---|---|---|---|---|---|---|
| | | | | Primate cone | Mouse cone | Primate rod | Mouse rod |
| Opsin decay rate const. | $\sigma$ | $s^{-1}$ | Free | 22 | 9.74 | 7.07 | 7.66 |
| PDE decay rate const. | $\phi$ | $s^{-1}$ | Constrained ($\varphi = \sigma$) | 22 | 9.74 | 7.07 | 7.66 |
| PDE dark activation rate | $\eta$ | $s^{-1}$ | Free | 2000 | 761 | 2.53 | 1.62 |
| Dark current | $I_D$ | pA | Measured | −240 to −428 | −41 to −80 | −19 to −37 | −16 to −24 |
| cGMP concentration in dark | $G_D$ | $\mu M$ | Derived | 28.7 to 35 | 15.9 to 20 | 12.1 to 15.0 | 11.4 to 13.1 |
| cGMP-to-current constant | $k$ | $pA\,\mu M^{-3}$ | Fixed | 0.01 | 0.01 | 0.01 | 0.01 |
| cGMP channel cooperativity | $n$ | Unitless | Fixed | 3 | 3 | 3 | 3 |
| $Ca^{2+}$ concentration in dark | $C_D$ | $\mu M$ | Fixed | 1 | 1 | 1 | 1 |
| $Ca^{2+}$ extrusion rate constant | $\beta$ | $s^{-1}$ | Fixed (rods) Free (cones) | 9 | 2.64 | 25 | 25 |
| Cooperativity of GC $Ca^{2+}$ dependence | $m$ | Unitless | Fixed | 4 | 4 | 4 | 4 |
| Affinity of GC $Ca^{2+}$ dependence | $K_{GC}$ | $\mu M$ | Free | 0.5 | 0.4 | 0.5 | 0.4 |
| Opsin gain | $\gamma$ | Unitless | Free | 10 | 10 | 4.2 | 8 |

that fit responses across stimuli. We identified model parameters by numerically minimizing the MSE between the model predictions and measured responses across stimuli (see Methods).

*Figure 1B* compares the predicted and measured responses of a mouse rod photoreceptor. Predicted and measured responses are not identical – for example the model predicts a faster initial rising phase of the flash response than that measured and underestimates the amplitude of flashes delivered shortly after a light step. Nonetheless, a common model specified by four free parameters ($\gamma$, $\sigma$, $\eta$, and $K_{GC}$) accounts for many aspects of the responses to this set of stimuli. These parameters were well constrained by fitting the measured responses: best-fit parameters varied by ~10% across different photoreceptors, and the MSE increased significantly for 10–20% changes in parameters (see Methods for a more detailed analysis).

To arrive at consensus models for rod transduction currents, we fit measured responses from multiple rods from either primate or mouse simultaneously. We first used the measured dark current and the previously measured relation between current and cGMP (*Equation 4* in *Figure 1*; *Rieke and Baylor, 1996*) to specify the cGMP concentration in darkness for each cell. We then allowed $\gamma$ to vary between cells to account for differences in sensitivity. The remaining parameters ($\sigma$, $\eta$, and $K_{GC}$) were constrained to have the same value across cells. We then numerically identified best values for these parameters by minimizing the MSE between model predictions and the set of recorded rod responses. These consensus models provide our best estimate of how an unrecorded cell of a given type will respond; these models provide the foundation for the manipulations of the photoreceptor responses in Figures 6–10. Consensus parameters for mouse and primate rods were very similar (see *Table 1*).

## Fitting cone phototransduction models

Due to the limited duration of the cone recordings, we focused on the variable mean noise stimulus. This stimulus was the most effective in constraining cone model parameters, likely because it strongly engages adaptation and thoroughly probes response dynamics.

We identified consensus cone models by fitting measured responses to the variable mean noise stimulus from multiple cones simultaneously, as described above for the rods. Four parameters ($\sigma$, $\eta$, $K_{GC}$, and $\beta$) were constrained to have the same value across cones, while $\gamma$ was allowed to vary across cells. Consensus parameters for mouse and primate cones differed substantially, reflecting the ~twofold faster kinetics of primate cone responses (flash response time-to-peak is ~35 ms in primate cones, and ~70 ms in mouse cones; see *Table 1*). For primate cones, performance using these consensus parameters was very similar to that from the parameters in *Angueyra et al., 2022*; for consistency we used these published parameters for our consensus model.

## Model performance

*Figure 2* tests the performance of the consensus models for both rod and cone photoreceptors. We focused on the variable mean noise stimulus since it probes responses to rapid stimulus variations and large changes in mean intensity that strongly engage photoreceptor nonlinearities. For each individual cell, we specified the dark cGMP concentration ($G_D$) using the measured dark current and allowed the overall gain $\gamma$ to vary; the remaining parameters were set to the consensus model values in *Table 1*. The resulting models (red) captured the measured responses (black) well.

For each recorded cell, the consensus models captured more than 80% of the variance of the responses (*Figure 2B*); on average these models captured ~90% of the variance. We compared the performance of the consensus models with that of models fit to each cell individually. Consensus models and models fit to individual cells performed similarly (*Figure 2B*). Models also generalized well across stimuli and to cells that did not contribute to the fits (*Figure 2—figure supplement 1*). This suggests that the consensus models can be used to predict responses of unrecorded cells.

*Figure 2* also shows predictions of a linear phototransduction model (blue) (see Methods). Rod and cone phototransduction currents depend linearly on light intensity for low-contrast inputs (e.g. *Baylor et al., 1984*; *Hass et al., 2015*; *Schnapf et al., 1990*). Hence, the linear model was fit to responses of the full model to low-contrast inputs at the mean intensity of the variable mean noise stimulus (25 R*/rod/s for rods and 22,000 R*/cone/s for cones; see Methods for model construction details); this means that the differences between the linear and full model responses can be directly attributed to nonlinearities in phototransduction. Linear predictions deviated from the measured responses

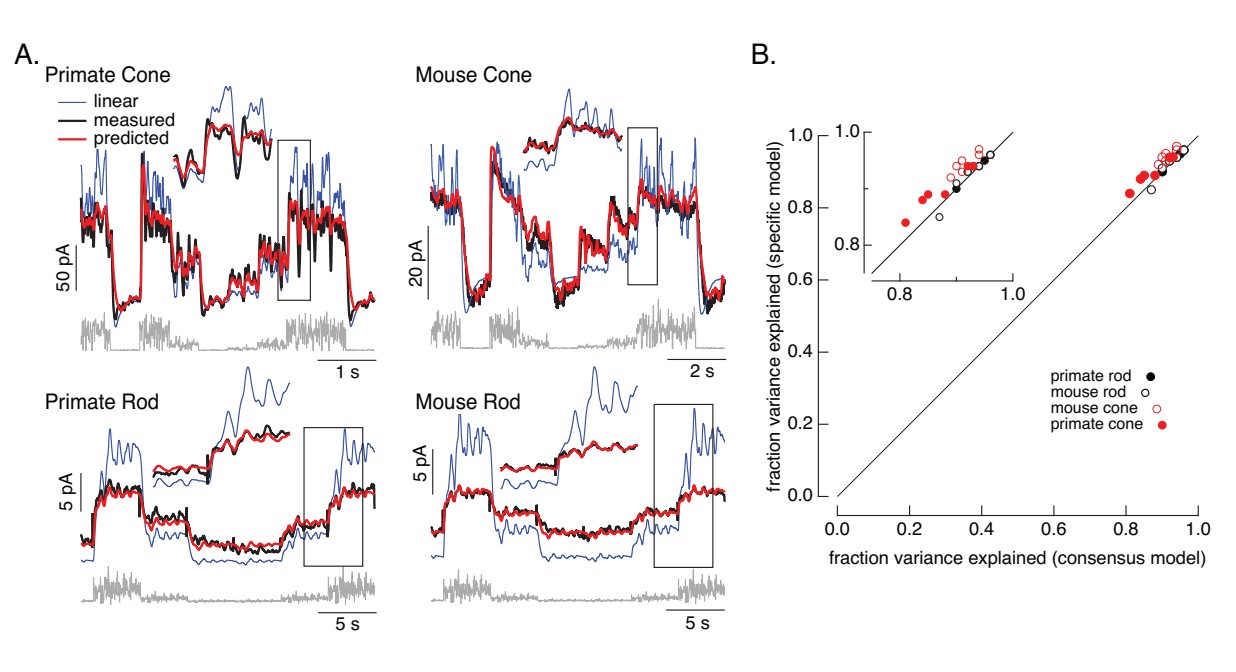

**Figure 2.** Photoreceptor model and fits. (**A**) Comparison of measured responses (black) with predictions of full (red) and linear model (blue) to variable mean noise stimulus (gray). Full model responses used consensus parameters from fitting responses of multiple cells of each type simultaneously, with the dark current and sensitivity allowed to vary between cells (see Methods and *Table 1*). The linear model was generated from fitting the low-contrast responses of the full model (see Methods). Insets expand regions in gray boxes. Linear model parameters (see *Equation 7*) were α = 0.31, $\tau_R$ = 10.6 ms, and $\tau_D$ = 23.6 ms for the primate cone, α = 0.031, $\tau_R$ = 31.6 ms, and $\tau_D$ = 56.7 ms for the mouse cone, α = 5.3, $\tau_R$ = 141 ms, and $\tau_D$ = 208 ms for the primate rod, and α = 4.7, $\tau_R$ = 115 ms, and $\tau_D$ = 185 ms for the mouse rod, (**B**) Fraction of variance explained for the full model fit to each cell individually (y-axis) plotted against that for the consensus model that has fixed parameters across cells except for the dark current and sensitivity. Means ± SDs were 0.90 ± 0.04 (specific model) and 0.87 ± 0.05 (consensus model) for six primate cones, 0.92 ± 0.02 and 0.94 ± 0.02 for six mouse cones, 0.94 ± 0.02, 0.94 ± 0.02 and 0.94 ± 0.02 for six primate rods and 0.93 ± 0.03 and 0.93 ± 0.03 for eight mouse rods. Data for Figures 1-5 is available at https://doi.org/10.5061/dryad.q2bvq83vg.

The online version of this article includes the following figure supplement(s) for figure 2:

**Figure supplement 1.** Tests of the ability of the model to generalize across stimuli and across cells.

**Figure supplement 2.** Full models systematically outperform linear models.

considerably more than did predictions of our biochemical model (*Figure 2—figure supplement 2*). Specifically, linear models overpredicted responses at high mean light levels, and underpredicted responses at low mean light levels. Some of these issues could be resolved by adding a free scale factor (e.g. in *Figure 2—figure supplement 2*) or a time-independent nonlinearity to the linear model (i.e. a linear–nonlinear model); however, responses to many stimuli show clear time-dependent nonlinearities that are not captured by linear–nonlinear models (e.g. Figure 6; see also *Angueyra et al., 2022*). The biochemical model, though not perfect, tracked the measured responses quite well.

*Figures 1 and 2* indicate that the known components of the phototransduction cascade, with appropriate parameters, can account for the full time course of the rod and cone photocurrents, including the nonlinearities in the photoreceptor responses. These models will not capture slow forms of adaptation that might become important, for example, at high light levels; for the range of stimuli used here, such slow adaptation contributes little to the photoreceptor currents (e.g. *Angueyra et al., 2022*). Our central goal was to use these models to design stimuli that permit predictable manipulations of the photoreceptor responses. Hence, more important than the accuracy of the forward model is the ability to predict stimuli the elicit desired responses. We directly evaluate these stimulus predictions below.

## Model inversion

In this section, we show that the model illustrated in *Figure 1A* can be exactly inverted to identify the stimulus that corresponds to a specific desired photoreceptor response. The resulting unique

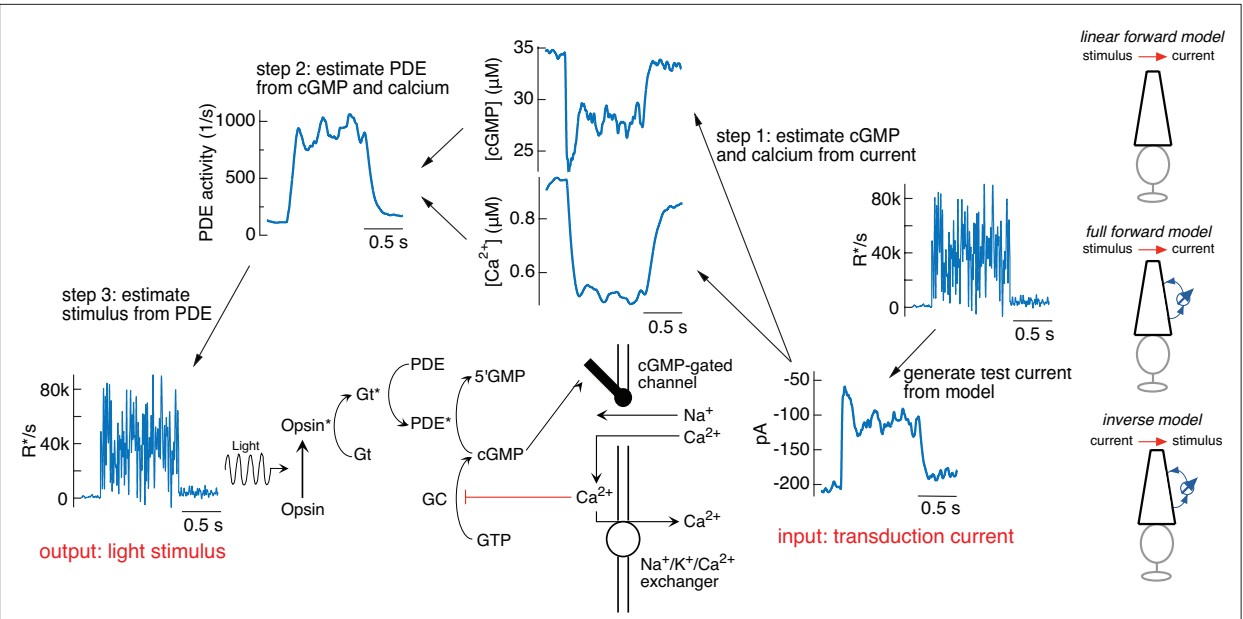

**Figure 3.** Steps in model inversion. A test current was generated from a variable mean noise stimulus using the full model (far right). Step 1 (right) converts this current into changes in cGMP and calcium using **Equations 4 and 5** (see **Figure 1**). Step 2 converts the time course of the cGMP and calcium into that of the phosphodiesterase (PDE) activity using **Equations 3 and 6**. Finally, step 3 converts the PDE to the stimulus using **Equations 1 and 2**. The estimated stimulus is identical to the initial stimulus because there is no added noise and the inversion process is exact. Icons at the far right depict different versions of the model used throughout the paper.

correspondence between photocurrent and stimulus applies to the entire stimulus waveform, including the mean intensity and modulations about the mean. In other words, the result of model inversion is an estimate of the full stimulus in real units (isomerizations per second) corresponding to the desired response.

Model inversion requires a one-to-one relationship between inputs and outputs – that is for every input there must be a unique output and vice versa. Mathematically, this holds for the phototransduction model summarized in **Figure 1A**: the model consists of linear elements (**Equations 1, 2, and 5**), static nonlinear elements (**Equations 4 and 6**), and a time-dependent nonlinear element (**Equation 3**). The linear elements can be described as linear filters and inverted by deconvolution. The static nonlinear elements consist of one-to-one mappings of inputs to outputs and can be inverted correspondingly. The time-dependent nonlinear component can be rearranged and solved analytically (see Methods). The result is an analytical relationship between the phototransduction current and the stimulus – in other words an inverse model that takes the photocurrent as input and identifies the corresponding stimulus.

**Figure 3** illustrates the steps in this procedure, using the current response generated by the forward model to the variable mean noise stimulus as an example 'target' (far right). The first step is to convert this target current response to corresponding time-varying concentrations of cGMP and calcium using **Equations 4 and 5** (step 1). In step 2, the time courses of cGMP and calcium are used to determine the time-varying PDE activity using **Equations 3 and 6**. In the final step (step 3), the PDE activity is used to determine the stimulus through **Equations 1 and 2**. In this case, as expected, we recover exactly the stimulus that we started with, confirming that mathematically the phototransduction model can be inverted.

The procedure illustrated in **Figure 3** is exact in the context of the phototransduction model – that is, given the architecture of the model, there is a one-to-one mapping of inputs to outputs. In practice, we would like to use this procedure to predict stimuli that will elicit specific photoreceptor responses. These model-based predictions could fail if the forward model does not accurately capture aspects of the real photoreceptor responses that are important for inversion. The model, for example, may capture some temporal frequencies of the response better than others, and this could limit the accuracy of the model inversion at those frequencies that are not captured well.

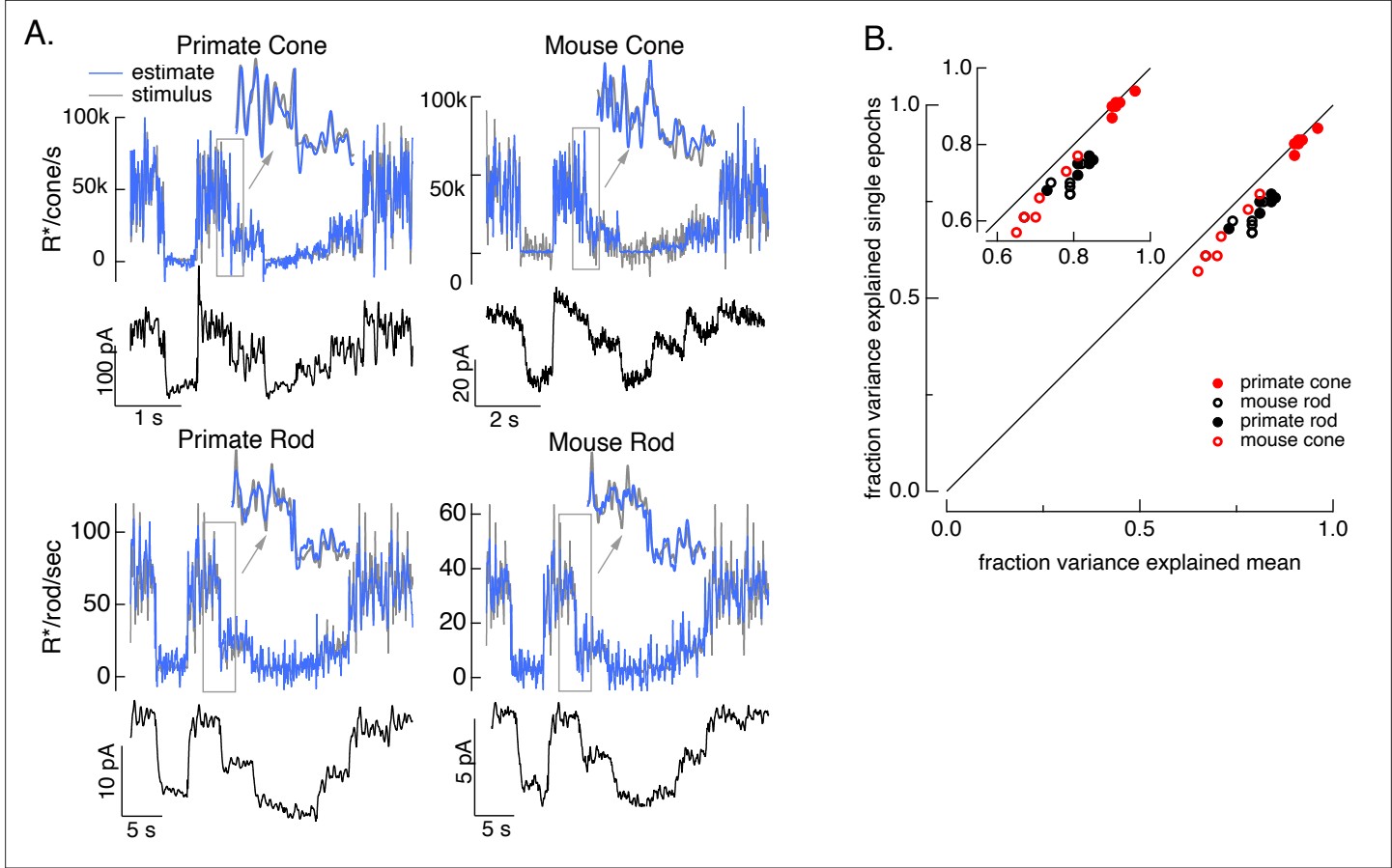

**Figure 4.** Test of model inversion based on measured responses. (**A**) Stimulus (gray in top panel), measured response (black in lower panels), and estimated stimulus (blue in top panels) calculated by using the measured response as input to the inverse model as in *Figure 3*. Estimates are able to recover both the periodic changes in mean intensity and the more rapid superimposed stimulus modulations (insets). (**B**) Variance explained for stimulus estimates based on the average response across multiple stimulus trials compared to that based on individual responses. Since the model captures only the deterministic part of the response, noise in the individual responses lowers the accuracy of the estimates and causes the points to fall below the unity line. This effect is modest but systematic. Means ± SDs were 0.91 ± 0.02 (single epochs) and 0.92 ± 0.02 (mean) for six primate cones, 0.66 ± 0.08 and 0.72 ± 0.06 for six mouse cones, 0.74 ± 0.03 and 0.81 ± 0.04 for six primate rods, and 0.68 ± 0.04 and 0.77 ± 0.05 for eight mouse rods.

To test how well the model inversion process works on real responses, we applied it to measured photoreceptor responses to the variable mean noise stimulus. Noise in the measured responses could cause the estimate to be better at some temporal frequencies than others; this is particularly true at high temporal frequencies that are poorly encoded by the photoreceptors. The limited ability to recover spatial information in deconvolution microscopy provides an example of how noise can limit model inversion even with a good forward model. Noise in deconvolution microscopy is controlled by choosing not to try to recover spatial frequencies at which the noise is too high. We took a similar approach: we limited the frequency content of the stimulus to temporal frequencies that we could reasonably expect to recover (0–60 Hz for primate cones, 0–30 Hz for mouse cones, 0–15 Hz for both primate and mouse rods) and required that the power spectrum of our estimate matched that of the true stimulus (see Methods). These constraints reduced high-frequency noise in the estimated stimuli. *Figure 4* shows stimulus estimates based on measured responses for each photoreceptor type. The estimates closely approximate the actual stimulus, including recovering rapid stimulus fluctuations (insets). For each photoreceptor type, stimulus estimates based on model inversion captured the majority of the stimulus variance (*Figure 4B*).

Variance explained, as in *Figure 4B*, mixes low temporal frequencies for which we might expect inversion to work well with high temporal frequencies for which it is expected to do less well. To explore the accuracy of the inversion across temporal frequencies, we compared the power spectrum of the stimulus with that of the residuals given by the difference between the stimulus and estimate

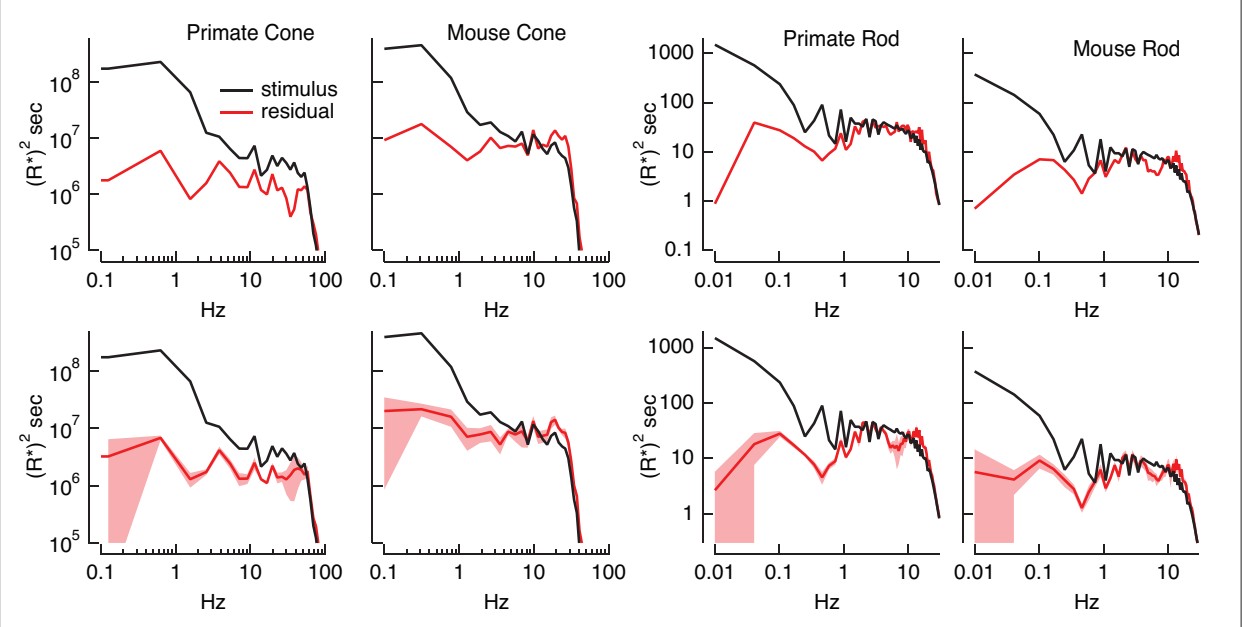

**Figure 5.** Errors in stimulus estimation. Power spectra of stimulus and residual (stimulus − estimate) for example primate and mouse rods and cones (top) and averages across cells (bottom; six primate cones, six mouse cones, six primate rods, eight mouse rods). The stimulus and residual power spectra diverge strongly at low frequencies where the estimates are close to the actual stimulus, and converge at higher frequencies when the estimate becomes poor. The convergence point depends on photoreceptor type, as expected for the different kinetics of rod and cone responses.

(*Figure 5*). At low temporal frequencies, stimulus power was much greater than that of the residuals, consistent with accurate stimulus estimation at these frequencies. This difference decreased with increasing frequency, and the two power spectra converged near 30 Hz for primate cones, 10 Hz for mouse cones, and 2–3 Hz for both primate and mouse rods. This provides an estimate of the frequency range over which stimulus inversion is possible for each photoreceptor type.

The ability to convert photoreceptor responses to input stimuli is useful in several ways (see Discussion). Our focus in the remainder of the Results is on using this approach to design stimuli that alter the photoreceptor responses in specific ways, such as negating the impact of adaptation and shaping response kinetics.

## Light-adaptation clamp examples

*Figure 6* illustrates how we can use the model inversion procedure to predictably manipulate photoreceptor responses, in this case using the cone model and (for simplicity) a sinusoidal stimulus. Unlike

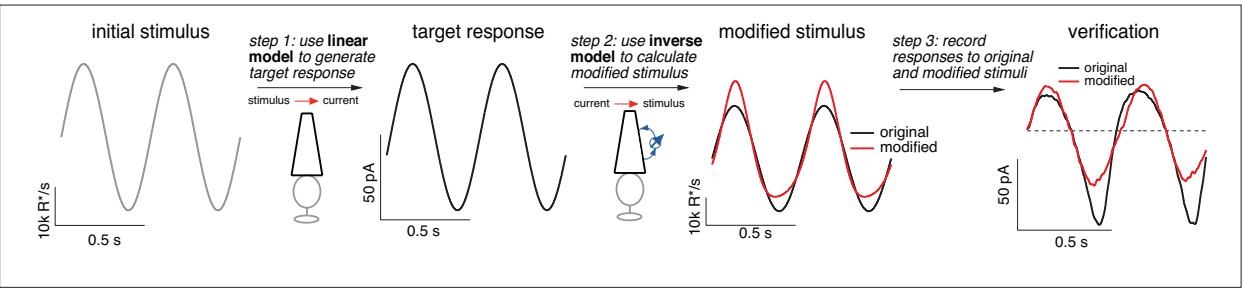

**Figure 6.** Light-adaptation clamp procedure. Starting with an initial stimulus (left), we generate a target or desired response (second panel from right). In this case, the target was chosen to be the response of a linear phototransduction model; for a sinusoidal input, a linear model produces a sinusoidal output. We use the (linear) target response as the input to the inverse phototransduction model and identify the stimulus required to elicit that response (red in third panel from left). Substantial stimulus modifications are required for the full model to produce a sinusoidal output. Finally, we confirm that the modified stimulus works as designed in direct recordings, in this case from a primate cone (right panel). Linear model parameters used to compute the target response (see *Equation 7*) were α = 0.31, $\tau_R$ = 10.6 ms, and $\tau_D$ = 23.6 ms.

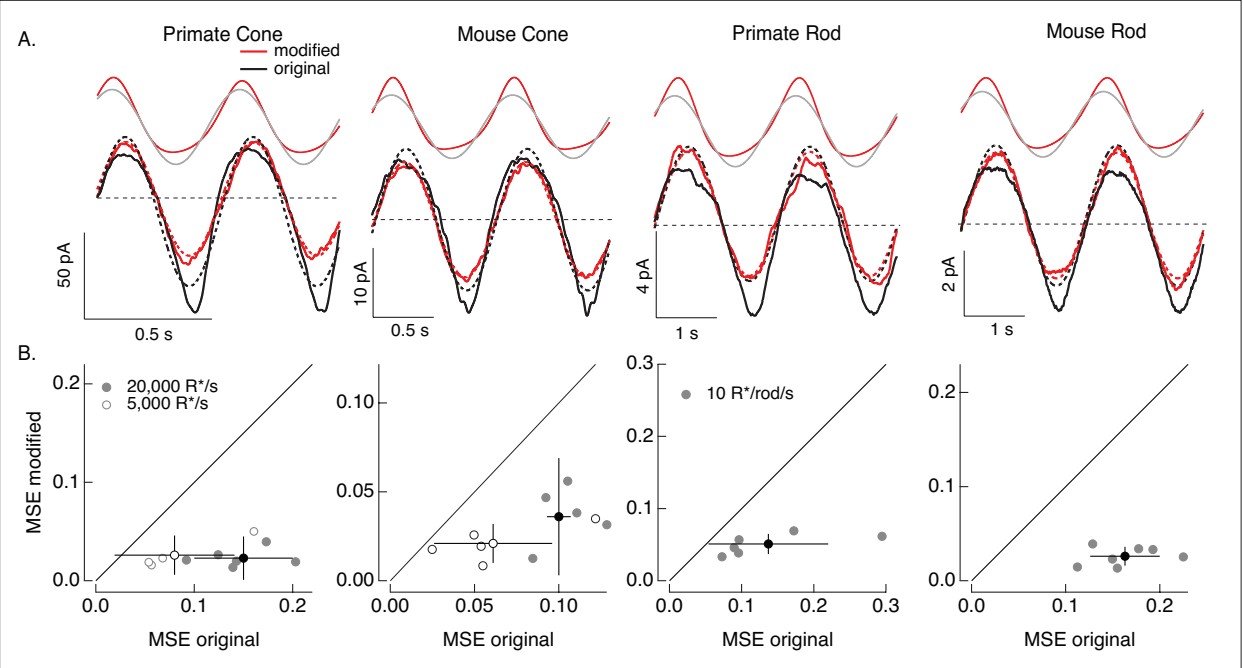

**Figure 7.** Compensating for increment/decrement asymmetries in responses to sinusoidal stimuli. (**A**) Photoreceptor responses to original sinusoidal stimuli (bottom, black) and modified stimuli (red). Dashed lines in the bottom panels are best-fitting sinusoids for reference. Original and modified stimuli are shown above the recorded responses. Linear model parameters used to compute the target responses (see *Equation 7*) were α = 0.31 (20,000 R*/s) and 1.65 (5000 R*/s), $\tau_R$ = 10.6 and 15.2 ms, and $\tau_D$ = 23.6 and 18.1 ms for the primate cone, α = 0.031 and 0.13, $\tau_R$ = 31.6 and 26.7 ms, and $\tau_D$ = 56.7 and 50.3 ms for the mouse cone, α = 5.3, $\tau_R$ = 141 ms, and $\tau_D$ = 208 ms for the primate rod, and α = 4.7, $\tau_R$ = 115 ms, and $\tau_D$ = 185 ms for the mouse rod. (**B**) Mean-squared error between measured responses and best-fitting sinusoids for modified stimuli plotted against that for original stimuli for each recorded cell. Points with error bars are means ± SDs. These values are 0.15 ± 0.05 (modified) and 0.02 ± 0.02 (original) at 20,000 R*/s and 0.08 ± 0.06 and 0.03 ± 0.02 at 5000 R*/s for six primate cones, 0.10 ± 0.01 and 0.04 ± 0.03 (20,000 R*/s) and 0.06 ± 0.03 and 0.02 ± 0.01 (5000 R*/s) for five mouse cones, 0.14 ± 0.08 and 0.05 ± 0.01 for six primate rods, and 0.16 ± 0.04 and 0.03 ± 0.01 for seven mouse rods.

*Figure 4* where we invert measured responses, we now start with a desired or target response. For the specific applications described below, the aim was to remove nonlinearities in photoreceptor responses and hence we generate the target response from a linear phototransduction model. As in *Figure 2*, the linear model used to generate the target response was obtained by fitting responses of the full phototransduction model to low-contrast noise stimuli at a specified mean light level (see Methods). The output of this linear model provides an estimate of how the photoreceptors would respond if we could eliminate their nonlinear properties. The response of the linear model to a sinusoidal stimulus is, as expected, another sinusoid (*Figure 6*, second panel from left). The response of real cones, and of the full cone model, deviates strongly from a sinusoid (black trace in the far right panel; *Angueyra et al., 2022*).

Adaptive nonlinearities in phototransduction cause the responses to sinusoidal light inputs to deviate from sinusoids. These time-dependent nonlinearities operate sufficiently rapidly to shape each cycle of the response, causing responses to decreases in light intensity to be larger than those to increases and causing the responses to light-to-dark transitions to be slower than those to dark-to-light transitions. To eliminate the impact of these nonlinear response properties, we seek a stimulus that will cause the response of the full model to match the target response – that is a stimulus that 'clamps' the cone response to the target. The model inversion process identifies such a stimulus directly (red trace in second panel from right). Our primary interest is not in the shape of the stimulus itself but instead in the response that the stimulus produces. The right panel in *Figure 6* compares measured cone responses to the original (black) and modified (red) stimuli; responses to the modified stimulus are considerably more sinusoidal than those to the original stimulus, as quantified in more detail below.

The approach outlined in *Figure 6* is exact in principle, but could fail due either to inadequacies of the model or to cell-to-cell variability in the photoreceptor responses. Hence, a key step is testing

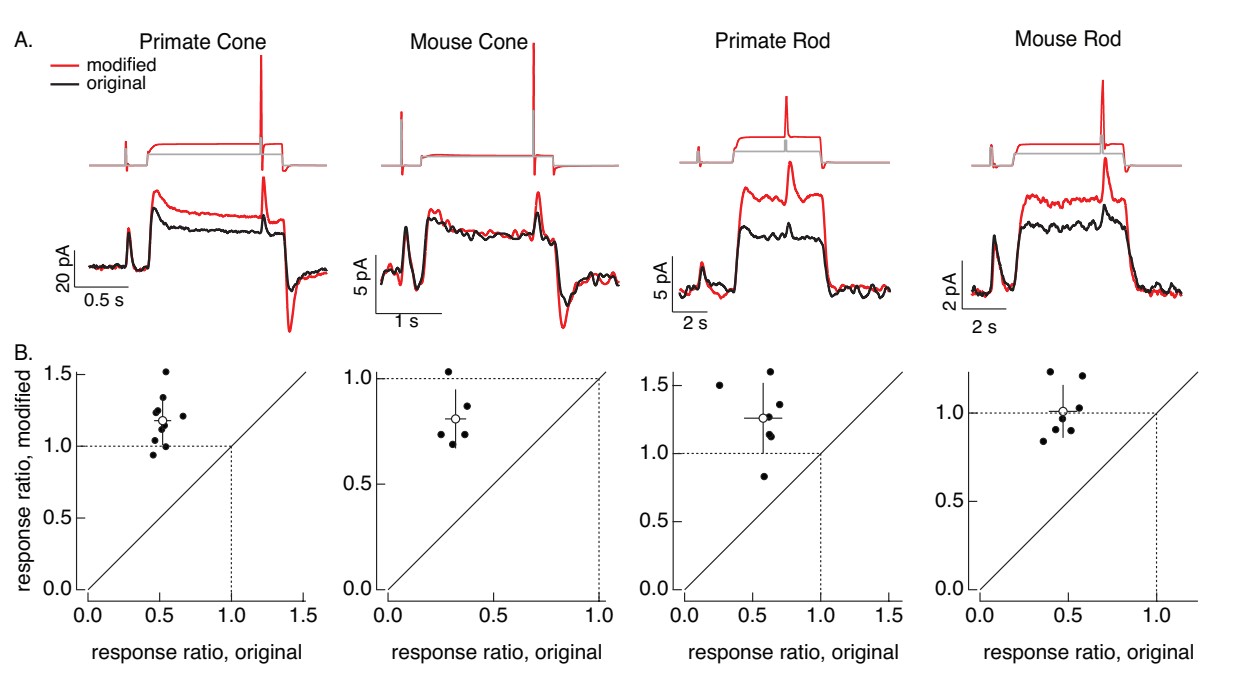

**Figure 8.** Compensating for adaptation produced by changes in mean light level. (**A**) Responses to a brief light flash delivered before and during a step in light intensity. For the original stimuli, flashes delivered before and during the step are identical, and the resulting responses decrease in amplitude ~twofold (summarized on x-axis of bottom panels). Red traces show responses to stimuli designed to compensate for the adaptation produced by the change in light intensity (following approach in **Figure 6**). Linear model parameters used to compute the target responses (see **Equation 7**) were $\alpha$ = 1.65, $\tau_R$ = 15.2 ms, and $\tau_D$ = 18.1 ms for the primate cone, $\alpha$ = 0.13, $\tau_R$ = 26.7 ms, and $\tau_D$ = 50.3 ms for the mouse cone, $\alpha$ = 5.3, $\tau_R$ = 141 ms, and $\tau_D$ = 208 ms for the primate rod, and $\alpha$ = 4.7, $\tau_R$ = 115 ms, and $\tau_D$ = 185 ms for the mouse rod. (**B**) Summary of gain changes (amplitude of response during the step divided by that of response prior to step) for responses to modified (y-axis) and original (x-axis) stimuli. Open circles show the mean ± SD; values are 1.2 ± 0.2 (modified) and 0.52 ± 0.06 (original) for 10 primate cones, 0.8 ± 0.1 and 0.32 ± 0.05 for 5 mouse cones, 1.3 ± 0.3 and 0.6 ± 0.1 for 7 primate rods, and 1.0 ± 0.1 and 0.47 ± 0.08 for 7 mouse rods.

whether the calculated stimuli produce the expected responses. **Figures 7–9** test this approach quantitatively for several stimulus manipulations.

## Sinusoidal stimuli

We start with the sinusoidal stimuli illustrated in **Figure 6**. As predicted by the model, responses of both rod and cone photoreceptors to high-contrast sinusoidal light inputs are strongly nonsinusoidal: responses to decreases in light intensity are larger than those to increases, and responses to dark-to-light transitions are faster than those to corresponding light-to-dark transitions (**Figure 7A**). These nonlinearities in the photoreceptor responses are clear from the substantial deviations between the measured responses and the best-fitting sinusoids (dashed lines in **Figure 7A**). These response asymmetries are important in interpreting responses of downstream visual neurons to similar stimuli. For example, asymmetries in signaling of On and Off ganglion cells are often attributed to differences in On and Off retinal circuits (e.g. **Stockman et al., 2014**), and the possibility that they arise at least in part in the photoreceptors themselves is rarely considered.

Responses to the modified stimuli (red traces in **Figure 7A**) were much closer to sinusoidal than responses to the original sinusoidal stimuli (black traces). Specifically, asymmetries between increment and decrement responses and light-to-dark vs dark-to-light transitions were substantially reduced for the modified stimuli. We quantified these deviations of the measured responses from a sinusoid for both original and modified stimuli by computing the MSE between the recorded responses and the best-fit sinusoid. Sinusoidal fits to responses to the modified stimuli had considerably lower MSE than those to the original stimuli in each rod and cone tested (**Figure 7B**). Neither the sinusoidal stimuli nor the specific cells that contribute to **Figure 7** were used in fitting the phototransduction models.

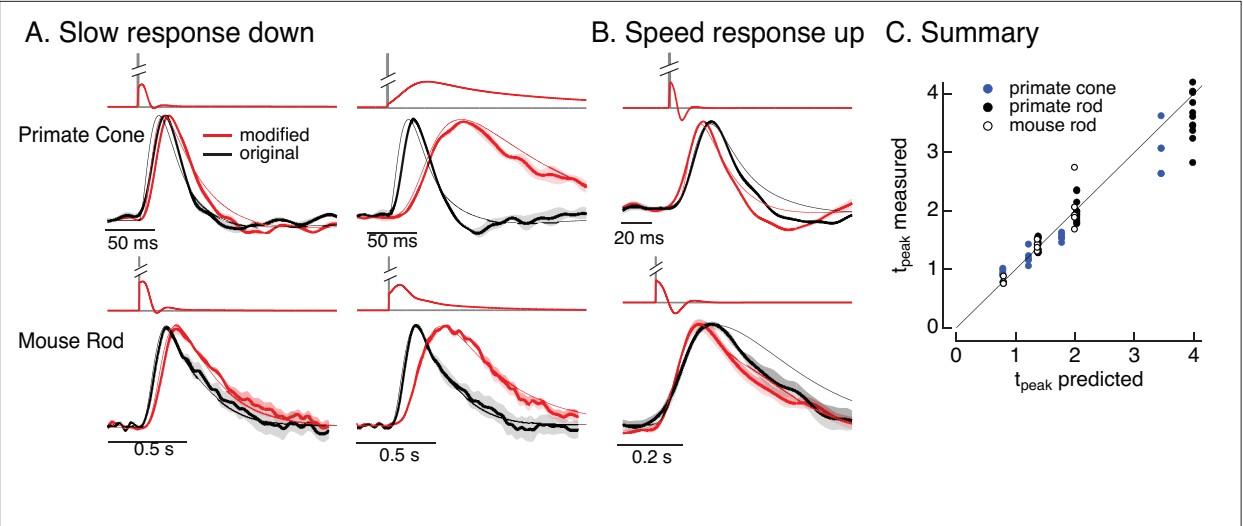

**Figure 9.** Manipulating kinetics of photoreceptor responses. (**A**) Responses of primate cones and mouse rods to brief flashes (black) and stimuli designed to slow down responses slightly (left) and more substantially (right) (red). Traces show the mean ± SEM (four cells for upper left, three for upper right, five for both bottom panels). Thin traces show the target responses used to generate the modified stimuli. (**B**) As in A for stimuli designed to speed up responses (measured traces show mean ± SEM for four cones in the top panel and three rods in the bottom panel). (**C**) Measured change in time-to-peak plotted against predicted change for a variety of manipulations of kinetics as in A and B. Linear model parameters used to compute the target responses (see *Equation 7*) were $\alpha = 1.65$, $\tau_R = 15.2$ ms, and $\tau_D = 18.1$ ms for the primate cone, $\alpha = 0.13$, $\tau_R = 26.7$ ms, and $\tau_D = 50.3$ ms for the mouse cone, $\alpha = 5.3$, $\tau_R = 141$ ms, and $\tau_D = 208$ ms for the primate rod, and $\alpha = 4.7$, $\tau_R = 115$ ms, and $\tau_D = 185$ ms for the mouse rod.

Hence, *Figure 7* tests the ability of the model-based inversion process to generalize across photoreceptors and across stimuli.

*Figure 7* demonstrates that the adaptive nonlinearities in phototransduction that distort responses to sinusoidal stimuli can largely be compensated for, resulting in near-sinusoidal responses. These or similar modified stimuli could in turn be used to test the contribution of nonlinearities in phototransduction to asymmetries in responses of On and Off retinal neurons to sinusoidal stimuli such as drifting or contrast-reversing gratings.

## Steps and flashes

As a second example of the light-adaptation clamp procedure, we generated stimuli that compensate for the adaptive changes in gain of photoreceptor responses produced by changes in mean light intensity. These gain changes cause responses to a fixed strength flash to become smaller and faster when the mean light intensity increases.

The control stimulus consisted of identical flashes delivered at two mean light levels. As expected, responses to this stimulus (black traces in *Figure 8A*) showed clear effects of adaptation. Specifically, the response to the flash decreased ~twofold at the higher mean light level (summarized in *Figure 8B*). We then passed this original stimulus through a linear model to estimate the photoreceptor response without adaptation. This linear prediction provided the target response in the stimulus identification process illustrated in *Figure 6*; the resulting modified stimuli are shown in red at the top of *Figure 8A*. None of the cells used to test these modified stimuli contributed to the model fitting, so this again tests the ability of the models to generalize across photoreceptors.

The red traces in *Figure 8A* (bottom) show responses to the stimuli designed to compensate for adaptation. Responses to the test flashes had similar amplitudes at the low and high mean light level. This similarity is held across cells (the ratio of the response amplitudes is plotted on the *y*-axes in *Figure 8B*). The change in gain produced by the background was reduced in all photoreceptor types. Responses to the modified stimuli in some cases (e.g. the mouse cones) show a systematic dependence on background, likely due to differences in sensitivity between the cells used to fit and test the model. Nonetheless, the modified stimuli compensated for much of the reduction in response gain produced by adaptation.

## Altering kinetics

As for adaptation, separating the contributions of phototransduction and post-transduction circuit mechanisms to the kinetics of responses of downstream cells is difficult. Hence, we sought to use the model inversion process to predictably alter the kinetics of the photoreceptor responses, which could help elucidate the impact of photoreceptor kinetics on downstream signaling. For example, compensating for the speeding of photoreceptor responses that occurs with increasing light intensity would isolate the kinetics of the post-transduction circuitry and determine if they change independently from the changes in kinetics of the photoreceptor responses.

Slowing down the kinetics is conceptually straightforward – we slow down the kinetics of the stimulus itself and that will also cause the response to slow down. We applied this approach to responses to brief flashes. To test if we could do this in a predictable manner, we slowed the linear responses to a brief flash by factors of 1.25–4 by scaling the time constants in *Equation 7*. We used the resulting responses as the targets for the model inversion as in *Figure 6*. We omitted mouse cones because these experiments required longer-lasting recordings with stable kinetics, which were difficult to achieve with these cells.

The top panels in *Figure 9A* show original (black) and modified (red) stimuli designed to slow down the photoreceptor response slightly (left) and more substantially (right). The bottom panels compare measured (thick traces) and predicted (thin traces) responses to the original flash and the modified stimuli. Changes in time-to-peak of the measured responses to the modified stimuli follow the predictions well for both rod and cone photoreceptors (summarized in *Figure 9C*).

Speeding up the responses is similarly straightforward in principle but more difficult in practice. Stimuli that speed up the response consist of a brief increment followed by a decrement – which together cause all but the initial rising phase of the response to cancel. In practice, this approach is limited because the contrast of the decrement cannot exceed 100%. Nonetheless, speeding up responses by 20–30% is possible, and stimuli predicted to achieve this are shown in *Figure 9B*. Measured responses sped as predicted, although the falling phase of the response, particularly for mouse rods, was faster than the predicted response. This again likely reflects differences in the specific cells used to fit and test the model. As for slowing responses, the time-to-peak of the measured responses closely follows predictions (*Figure 9C*).

## Impact of cone adaptation on ganglion cell responses

The overall goal of this project is to separate the contributions of phototransduction and post-transduction circuit mechanisms to downstream visual signaling. Here, we show two examples of how the ability to manipulate photoreceptor responses can reveal adaptational mechanisms at different locations in retinal circuits.

First, we compare responses of primate cones and downstream retinal neurons to the step plus flash stimulus used in *Figure 8*. *Figure 10A* shows responses of a primate cone, a horizontal cell, and an On parasol ganglion cell to the original step plus flash stimulus (black) and the modified stimulus that negates cone adaptation (red). The stimuli used for the three cells illustrated were identical. As shown in *Figure 8*, the modified stimulus effectively negates adaptation in the cone transduction currents, and the responses to the cone responses to the two flashes are similar. Negating adaptation in the cones also eliminated most or all of the adaptation in downstream responses (summarized in *Figure 10B*). Ratios exceeding one are likely due to differences in sensitivity between the cells used to fit and test the model. Previous work reached a similar conclusion by quantitatively comparing adaptation measured in cones and retinal ganglion cells (*Dunn et al., 2007*). Negating cone adaptation provides an alternative approach to this issue and similar questions about the contributions of photoreceptor and post-photoreceptor adaptation.

Second, we recorded responses of primate horizontal cells and cone bipolar cells to sinusoidal stimuli and stimuli designed to produce sinusoidal cone transduction currents (as in *Figures 6 and 7*). Horizontal cell responses to both stimuli showed clear nonlinearities (*Figure 10C*, left); specifically, responses to the increment and decrement phases of the stimuli were highly asymmetric. The depolarizing and hyperpolarizing phases of the responses, however, appeared more symmetrical for the modified stimuli (red). The change in shape can be visualized by plotting the trajectories of one cycle of the responses vs the stimulus (*Figure 10C*, right). For the original sinusoid, the depolarizing (a→b→c) and hyperpolarizing (c-→d→a) response phases differed considerably. These differences were much

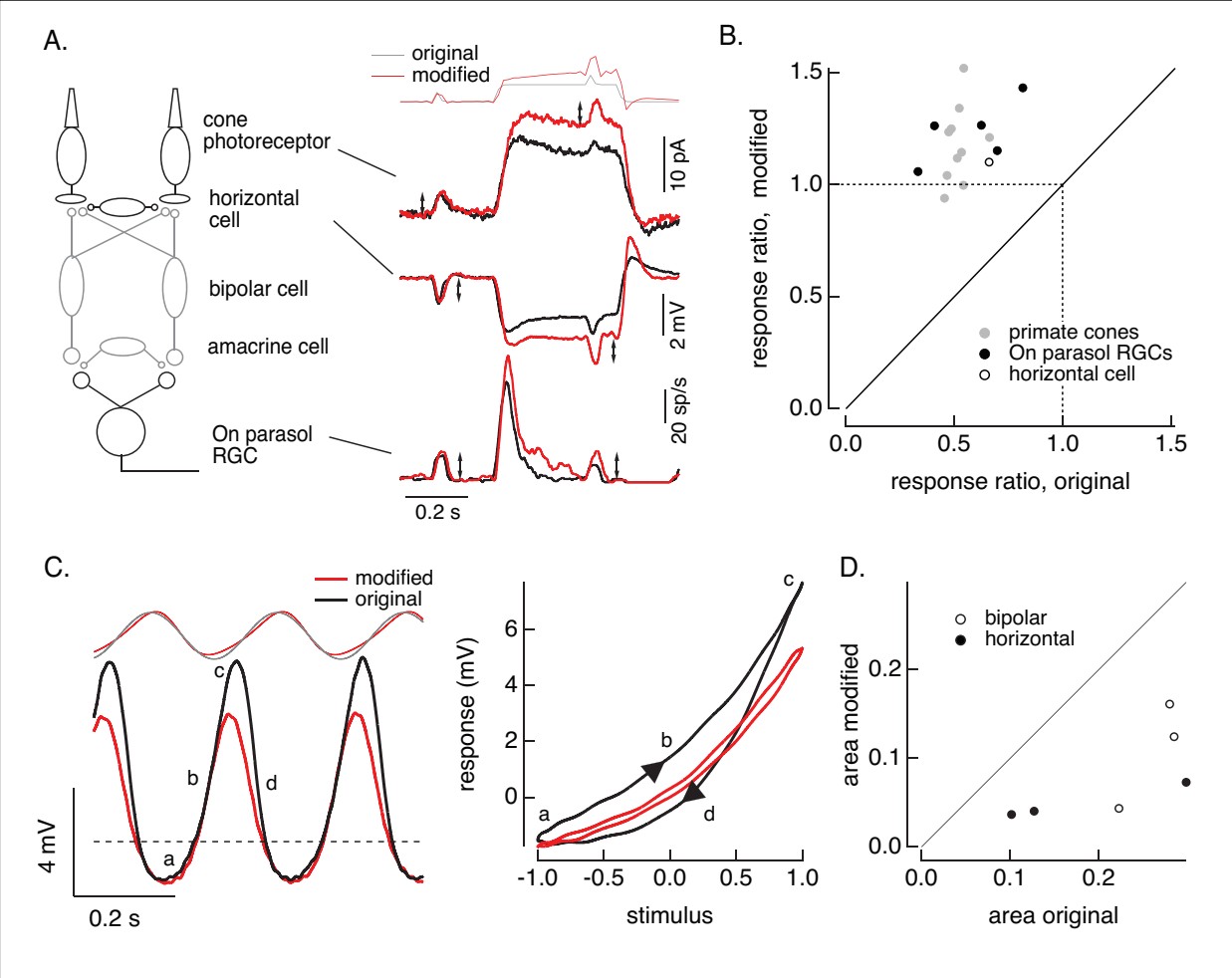

**Figure 10.** Cone and post-cone adaptation. (**A**) Left: schematic of retinal circuit. Right: responses of a cone (top), horizontal cell (middle), and On parasol ganglion cell (bottom) to the step and flash protocol for both original and modified stimuli. Stimuli were delivered from a computer monitor at a refresh rate of 60 Hz and hence appear different from those in *Figure 8*. (**B**) Summary of experiments like those in A, plotting the change in gain for the modified stimuli (the ratio of the amplitude of the response to the flash during the step that before) against that for the original stimuli. Means ± SDs were 1.2 ± 0.2 (modified) 0.52 ± 0.06 (original) for 10 cones and 1.2 ± 0.1 and 0.6 ± 0.2 for 5 parasol RGCs. (**C**) Left: horizontal cell responses to a sinusoidal stimulus (black) and stimulus modified to generate sinusoidal cone responses (red, see also *Figures 6 and 7*). Right: trajectories of responses to one stimulus cycle. The response to the original stimulus follows different trajectories during the depolarizing (a→b→c) phase compared to the hyperpolarizing (c→d→a) phase. (**D**) Summary of results from three horizontal cells and three cone bipolar cells. Time-dependent nonlinearities were measured from the area enclosed by response trajectories as in C. Means ± SDs were 0.08 ± 0.05 (area modified) and 0.22 ± 0.08 (area original) across all six cells. Linear model parameters used to compute the target responses (see *Equation 7*) were α = 1.65, $\tau_R$ = 15.2 ms, and $\tau_D$ = 18.1 ms.

smaller for the modified stimuli. This difference in response shape held across cells (*Figure 10D*). In this case, compensating for nonlinearities in cone phototransduction reveals an additional, largely time-independent, nonlinear process shaping horizontal and cone bipolar responses. Furthermore, it disentangles contributions of phototransduction and post-transduction processes to time-dependent and time-independent nonlinearities in the horizontal and bipolar responses.

*Figure 10* shows two relatively simple examples of how the stimulus design approach developed here can be used. We return to other possible applications in the Discussion.

## Discussion

Rod and cone photoreceptors play an essential role in the responses of downstream visual neurons and visual perception. Refining this picture to identify the specific contributions of photoreceptors and post-photoreceptor circuitry to the computations that underlie vision has been difficult. Here, we have created

and tested a tool that allows predictable manipulations of the photoreceptor responses to reveal their role in downstream signaling. Direct tests indicate that this tool is effective, and that it generalizes well across photoreceptors and stimuli. We envision this being useful to causally test how responses of downstream cells or perception are shaped by specific aspects of the photoreceptor responses such as adaptation. We have described in Methods procedures that should help translate this tool to other laboratories.

## Limitations

### Model validation

The procedure that we use to invert the phototransduction cascade is only as good as the model upon which it is based. Thus, a focus of the experiments presented here was to test a range of manipulations for stimuli and photoreceptors that did not contribute to the model fits. We did this by fitting consensus models to recordings from one set of photoreceptors of a given type, and then testing the stimuli generated by these models on different photoreceptors from different retinas. Discrepancies between the predicted and measured responses to these new stimuli reflect both failures of the model inversion and cell-to-cell variability. Systematic differences between predicted and measured responses were relatively small, and most of the differences appear to reflect variability between cells (e.g. due to cell-to-cell differences in sensitivity).

### Focus on phototransduction

Our model describes the relationship between light inputs and phototransduction currents. We restricted our consideration to photocurrents rather than photoreceptor synaptic output because the latter is shaped by several mechanisms (e.g. electrical coupling between photoreceptors *Copenhagen and Owen, 1976*; *Detwiler and Hodgkin, 1979*; *Schwartz, 1976* and horizontal feedback *Baylor et al., 1971*) that are less well understood than phototransduction. A similar approach could be extended beyond phototransduction to include electrical properties of the photoreceptor inner segment and photoreceptor output synapses when additional quantitative information about these mechanisms is available. The models we develop here will be important in such future work, as they will allow isolation of mechanisms operating after phototransduction (e.g. in *Figure 10C*).

### Photopigment bleaching

The model that we used does not consider photopigment bleaching and regeneration via the pigment epithelium. We omitted these aspects of photoreceptor function because the experiments to which we fit and test the model require that we remove the pigment epithelium. We restricted total light exposure in these experiments to minimize bleaching, and this limits the range of light levels over which the model is applicable. This is less of a concern for rods, where rod saturation occurs for light levels at which a small fraction of the rod pigment is bleached, and correspondingly our models cover most or all of the range of rod signals. For cones, the model is limited to light levels <50,000 R*/s (similar to or a bit brighter than typical indoor lighting conditions).

### Speeding vs slowing responses

Not all manipulations of the photoreceptor responses are equally easily achieved. Speeding responses is particularly difficult because it requires increasing the amplitude of high temporal frequencies, and the ability to do that is limited by the requirement that light intensities not assume negative values. This means that low-contrast stimuli can be sped more than high-contrast stimuli.

### Calibrations

The accuracy of the model predictions depends critically on accurate light calibrations. We have detailed our procedure in Methods and supplied our calibration code (https://github.com/chrischen2/photoreceptorLinearization, copy archived at *Chen, 2024*). With consistent calibrations, the approaches we use here should translate directly across laboratories and to in vivo physiological or perceptual studies.

## Applications

### 'Front end' for encoding models for downstream responses and perception

Two broad classes of encoding models have been used to describe responses of retinal ganglion cells and cells in downstream visual areas. Empirical models take light stimuli as inputs and convert these,

through a series of linear and nonlinear elements, to predicted responses (*Chichilnisky, 2001*; *Pillow et al., 2008*). Time-dependent nonlinearities are particularly hard to capture in such models, and few existing models account for them. CNN models similarly take light input and convert it, via the learned CNN weights, to predicted neural responses (*McIntosh et al., 2016*; *Turner et al., 2019*). Like empirical models, time-dependent nonlinearities such as adaptation are generally not well described by such models. As a result, these models work best when stimulus parameters such as mean light intensity do not change.

The photoreceptor models described here convert light inputs to photoreceptor responses, capturing time-dependent nonlinearities in this process. Empirical or CNN models could then be used to describe the conversion of photoreceptor signals to downstream neural responses. This architecture – a phototransduction model front end followed by an empirical model – should decrease the demand placed on the empirical model and improve the ability to capture responses to stimuli that strongly engage photoreceptor adaptation. Hybrid models of this type indeed show improved performance in predicting retinal ganglion cell responses across light levels (*Idrees et al., 2024*). Models for visual perception could similarly incorporate a photoreceptor front end, and by doing so directly test which aspects of perception can be explained by phototransduction and which are due to downstream processing.

## 'Back end' to decoding models

Approaches to decode neural responses and estimate stimulus properties could also benefit from the photoreceptor models. Current decoding approaches either empirically fit response–stimulus relationships or invert encoding models to compute the likelihood of particular stimuli given the neural response (*Bialek et al., 1991*; *Brackbill et al., 2020*; *Wu et al., 2024*). In either case, decoding in the context of stimuli that strongly engage time-dependent nonlinearities has proven difficult.

Incorporating the inverse photoreceptor model into decoding approaches should improve decoding performance. Specifically, existing decoding approaches could be used to estimate the photoreceptor signal from downstream neural responses (e.g. those of retinal ganglion cells), and then the inverse phototransduction model would convert the estimated photoreceptor signals to estimated stimuli. This again could decrease the demand placed on the decoding model. For example, under conditions in which much of retinal adaptation is largely accounted for by adaptation in the photoreceptors, the model used to estimate photoreceptor responses from retinal ganglion cell responses would not need to incorporate adaptive nonlinearities. Instead, this model could be fixed even as mean light levels change, and the inverse photoreceptor model would account for photoreceptor adaptation. A ganglion cell→photoreceptor decoding model should be simpler to fit and hence perform better than a full ganglion cell→stimulus model.

## Complementing genetic approaches

Genetic manipulations provide another approach to alter photoreceptor responses and characterize the impact on responses of downstream visual neurons or behavior. Such approaches have the advantage of associating specific molecular components with alterations in vision – for example the role of arrestin (*Burns et al., 2006*; *Xu et al., 1997*) and rhodopsin kinase (*Arshavsky, 2002*; *Wilden et al., 1986*; *Zhao et al., 1995*) in specific forms of stationary night blindness (*Dryja, 2000*; *Zeitz et al., 2014*). But interpretation of these genetic manipulations can be limited by compensatory changes.

The approach that we introduce here complements genetic manipulations in at least two ways. First, it could be used in conjunction with genetic manipulations – for example to restore normal kinetics in photoreceptors in which the genetic alteration changes response kinetics. Second, the stimulus design approach provides an alternative that provides less detailed mechanistic information, but which allows more specific functional manipulations to be made and does not require genetic access.

## Manipulations of phototransduction currents and identifying nonlinear circuit properties

The models that we introduce here provide a new tool to causally test the impact of alterations in photoreceptor responses on downstream responses and perception. For example, asymmetries in how increments and decrements in light intensity are processed have been well studied, including

responses in retina, cortex, and perception (*Bowen et al., 1989*; *Lu and Sperling, 2012*). Parallel On and Off visual pathways are initiated at the cone output synapse, and asymmetries in downstream responses or perception are often attributed to asymmetries in the On and Off retinal circuits conveying photoreceptor signals to the ganglion cells (e.g. *Stockman et al., 2014*). The implicit assumption is that photoreceptor inputs to On and Off retinal circuits are symmetrical. While this is likely the case for low-contrast stimuli, rapid adaptation in phototransduction means that high-contrast stimuli will often not produce symmetric input to those circuits, and On/Off asymmetries may originate at least in part from asymmetric photoreceptor responses to light increments and decrements (*Angueyra et al., 2022*; *Clark et al., 2013*; *Endeman and Kamermans, 2010*). The ability to invert the phototransduction model permits the design of stimuli that minimize such asymmetries in the photoreceptor responses to test how they contribute to On/Off differences in downstream responses (e.g. *Yu et al., 2022*).

More generally, the ability to shape photoreceptor responses in predictable ways provides a needed tool to isolate the effects of photoreceptor and post-photoreceptor circuits to shaping responses of downstream neurons and perception. Nonlinear circuit properties are a particular challenge since their impact depends on the input stimulus, yet they are also particularly important since they account for the bulk of interesting computation. Advances in modeling, particularly CNN-based approaches, can be used to fit circuit outputs and reveal how specific computations may be implemented. But such approaches rarely identify a unique explanation. Another approach is to record from each of the relevant circuit elements. This is technically challenging. Manipulating photoreceptor responses provides another tool, as illustrated in *Figure 10*. A key feature of this tool is the ability of our photoreceptor models to generalize across stimuli, including accounting for nonlinear properties of the photoreceptor responses to stimuli that did not contribute to the model directly.

## Methods

**Key resources table**

| Reagent type (species) or resource | Designation | Source or reference | Identifiers | Additional information |
|---|---|---|---|---|
| Biological sample (*Mus musculus*) | 129S1/SvlmJ | The Jackson Laboratory | RRID:IMSR_JAX:002448 | |
| Biological sample (*Mus musculus*) | C57BL/6J | The Jackson Laboratory | RRID:IMSR_JAX:000664 | |
| Biological sample (*Macaca fascicularis*) | | Washington Regional Primate Research Center | N/A | |
| Biological sample (*Macaca nemenstrina*) | | Washington Regional Primate Research Center | N/A | |
| Chemical compound, drug | Ames | MilliporeSigma | Cat #A1420 | |
| Software, algorithms | MATLAB | Mathworks | https://www.mathworks.com/products/matlab.html | |
| Software, algorithms | Stage | Stage-VSS | https://stage-vss.github.io | |
| Software, algorithms | Symphony | Symphony-DS | http://symphony-das.github.io | |
| Software, algorithms | Igor Pro | Wavemetrics | N/A | |

## Recordings

We performed electrophysiological recordings from primate (*Macaca fascicularis*, *nemestrina*, and *mulatta*, either sex, 2–20 years) and mouse retina (C57/BL6 or sv-129, either sex, 1–12 months) in accordance with the University of Washington Institutional Animal Care and Use Committee. Primate retinas were obtained through the Tissue Distribution Program of the University of Washington Regional Primate Research Center. Primate recordings were all at >20° eccentricity.

Rod responses were recorded with suction electrodes (*Field and Rieke, 2002*). These recordings were sufficiently stable that we could record responses to all three test stimuli in *Figure 1B*. Data were collected from any primate rod with a dark current exceeding 18 pA and any mouse rod with a dark current exceeding 15 pA. Periodic standardized flashes tested for changes in kinetics or amplitude of

the light response, and recordings were terminated if such changes were apparent. Most recordings were stable for at least 10–15 min before changes in kinetics were apparent, and all cells meeting this criterion are included in our analyses.

Cones were recorded with whole-cell patch clamp techniques in slice (mouse; *Ingram et al., 2019*) and whole-mount preparations (primate; *Angueyra and Rieke, 2013*). We focused on M and S cones in mouse and L and M cones in primate. Data were collected and analyzed from any primate cone with a dark current exceeding 200 pA and any mouse cone with a dark current exceeding 40 pA. Cone responses in these recordings run down quickly due to intracellular dialysis of the cell; hence, responses to only one of the stimuli in *Figure 1B* were recorded from each cone. All data reported here were collected within 2–3 min of patch rupture and the onset of intracellular dialysis.

## Light calibrations

Optical power was measured at the preparation with a calibrated power meter (Graseby). Power readings were converted to isomerizations per second (R*/s) using the photoreceptor spectral sensitivities, the LED spectral outputs, and collecting areas of rods and cones (1 and 0.37 $\mu m^2$ for primates, 0.5 and 0.2 $\mu m^2$ for mouse). This conversion proceeded in three steps: (1) the total calibrated power was converted to a power density ($W/\mu m^2$) using the size of the illuminated area in the microscope image plane; (2) LED emission spectra were scaled such that their integrals matched the calibrated power density; (3) the resulting calibrated LED emission spectra were converted to R*/s by taking the dot product with the photoreceptor spectral sensitivities (*Baylor et al., 1987*; *Nikonov et al., 2006*) and scaling by the collecting area (https://github.com/chrischen2/photoreceptorLinearization, copy archived at *Chen, 2024*).

Light stimuli from the LEDs were focused on the preparation via the microscope condenser or objective. Stimuli uniformly illuminated a 600-μm diameter spot. LED outputs were carefully checked for linearity. For rod recordings, we used an LED with peak output at 470 nm, near the peak of the rod spectral sensitivity. For cone recordings, we used an LED with peak output at 405 nm, which produced near-equal (within 10%) activation of L and M cones in primate and S and M cones in mouse. *Figure 10* used an OLED computer monitor (eMagin) to deliver stimuli; the monitor outputs were calibrated as above for the LEDs.

## Forward model

The phototransduction cascade was modeled with the set of differential equations illustrated in *Figure 1A*. This follows previous modeling work in both rods and cones (*Angueyra et al., 2022*; *Nikonov et al., 2000*; *Pugh and Lamb, 1993*; *Rieke and Baylor, 1998a*; *Younger et al., 1996*). Our model has 11 parameters. In the first step of the model, the light stimulus (Stim) activates opsin molecules, converting inactive opsin (*R*) to active opsin (*R\**). Active opsin decays with a rate constant $\sigma$ (*Figure 1A*).:

$$dR\left(t\right)/dt = \gamma Stim\left(t\right) - \sigma R\left(t\right) \tag{1}$$

Next, PDE molecules are activated by active opsin molecules via the G-protein transducin; the resulting PDE activity (*P*) decays with a rate constant $\Phi$ (*Figure 1A*):

$$dP\left(t\right)/dt = R\left(t\right) - \Phi P\left(t\right) + \eta \tag{2}$$

We assume that the delay caused by transducin activation is negligible and hence omit this step for simplicity. The opsin decay rate $\sigma$ and the PDE decay rate $\Phi$ were interchangeable (*Pugh and Lamb, 1993*), and the model output depended only on the smaller of these (i.e. the slower process). Hence, $\sigma$ and $\Phi$ were constrained to be equal.

The concentration of cGMP in the outer segment (*G*) is determined by the balance of cGMP hydrolysis mediated by PDE and synthesis (*S*) through guanylate cyclase (*GC*) (*Figure 1A*):

$$dG\left(t\right)/dt = S\left(t\right) - P\left(t\right)G\left(t\right) \tag{3}$$

For physiological conditions, the outer segment current (*I*) through cGMP-gated channels depends on a power of the cGMP concentration (*Figure 1A*; *Rieke and Baylor, 1996*):

$$I(t) = kG^n(t) \tag{4}$$

Calcium enters the outer segment through open cGMP channels and is removed by an exchange protein, with a rate constant $\beta$ (**Figure 1A**):

$$dC(t)/dt = qI(t) - \beta C(t) \tag{5}$$

The calcium concentration ($C$) regulates the rate of cGMP synthesis ($S$); this dependence is modeled as a Hill curve (**Figure 1A**):

$$S = \frac{S_{max}}{1 + (C/K_{GC})^m} \tag{6}$$

Two model parameters were fixed by steady-state conditions and prior measurements. First, the constant $q$ that relates current and changes in calcium can be expressed in terms of the dark calcium concentration, dark current, and rate constant $\beta$ for calcium extrusion:

$$q = \frac{\beta \cdot C_D}{I_D}$$

Second, the maximal cyclase rate ($S_{max}$) can be written in terms of $\Phi$, $\eta$, $K_{GC}$, $m$, and the dark calcium ($C_D$) and cGMP concentrations ($G_D$):

$$S_{max} = \frac{G_D \cdot \eta}{\Phi \cdot \left(1 + (C_D/K_{GC})^m\right)}$$

We fixed the constants $k$ and $n$ that determine the relation between cGMP and current at values measured for rods (e.g. **Rieke and Baylor, 1996**); modest changes in these parameters produced little or no change in model performance due to compensation by other model parameters. Using these constants, the cGMP concentration in darkness ($G_D$) was estimated from the measured dark current ($I_D$) using **Equation 4** (e.g. the response to a saturating light flash) for each recorded cell. We also fixed $m$ for both rods and cones and $\beta$ for rods based on previous measurements (**Field and Rieke, 2002**; **Koutalos et al., 1995a**; **Rieke and Baylor, 1996**). We assumed a dark calcium concentration ($C_D$) of 1 μM (**Gray-Keller and Detwiler, 1994**; **Sampath et al., 1999**); the model was insensitive to this value, as the dependence of the synthesis rate on calcium ($K_{GC}$) compensated for any change.

This left a total of four free parameters for rod models ($\gamma$, $\eta$, $\sigma$, $K_{GC}$), and five for cone models ($\gamma$, $\eta$, $\beta$, $\sigma$, $K_{GC}$). These parameters were determined by minimizing the mean-squared difference between the measured responses and model predictions while the remaining model parameters were held fixed. Some combinations of model parameters can trade for each other, resulting in similar model performance (e.g. $C_D$ and $K_{GC}$). Our goal here was to identify a model that captures photoreceptor responses across a range of stimuli, rather than a model in which the fit parameters were predictions of the underlying biochemical parameters.

We identified optimal values for the free parameters (see **Table 1**) using the MATLAB fminsearch routine, which employs the Nelder–Mead simplex algorithm. The time step in these calculations was 0.1 ms for cones and 1 ms for rods. The objective function minimized the MSE between the measured and predicted responses. The goodness of fit was assessed using the fraction of variance explained, calculated as $1 - (SSE/SST)$, where $SSE$ is the sum of squared errors between the model and the data, and $SST$ is the total sum of squares (variance) of the data. The model's performance was validated using independent datasets.

Parameters of consensus models were determined by simultaneously fitting measured responses across all recorded cells of a given type. To evaluate the model's ability to generalize across different cells and stimuli, we fixed all parameters except for $G_D$ and $\gamma$. $G_D$ was determined from the measured dark current ($I_D$) for each individual photoreceptor. $\gamma$ was allowed to vary between cells to account for differences in sensitivity, while the remaining parameters were constrained to be identical. For example, consensus parameters for a set of five cones would have common free parameters of $\eta$, $\beta$, $\sigma$, $K_{GC}$ and five individual $\gamma$s (one for each cell). Consensus parameters are given in **Table 1**. Fitted $\gamma$ values were 8 ± 3 (mean ± SD) for primate cones, 3 ± 1 for mouse cones, 5.3 ± 0.7 for primate rods, and 7.4 ± 2.4 for mouse rods.

**Table 2.** Parameter variations that double the mean-squared error of the fits.
Error was measured while a single parameter was systematically varied; other parameters were held fixed at the consensus model values.

| Parameter | Symbol | Units | Values: optimal [lower, upper] | | | |
| --- | --- | --- | --- | --- | --- | --- |
| | | | Primate cone | Mouse cone | Primate rod | Mouse rod |
| Opsin decay rate const. | $\sigma$ | s$^{-1}$ | 22 [14, 34] | 9.74 [7.3, 13.1] | 7.07 [5.8, 8.8] | 7.66 [6.1, 9.3] |
| PDE dark activation rate | $\eta$ | s$^{-1}$ | 2000 [1300, 3300] | 761 [494, 1218] | 2.53 [2.1, 3.2] | 1.62 [1.3, 2.1] |
| Ca$^{2+}$ extrusion rate const. | $\beta$ | s$^{-1}$ | 9 [5, 40] | 2.64 [1.58, 5.7] | N.A. | N.A. |
| Ca$^{2+}$ GC affinity | $K_{GC}$ | $\mu$M | 0.5 [0, 0.6] | 0.4 [0.26 0.46] | 0.5 [0.44, 0.53] | 0.4 [0.32, 0.44] |

We evaluated the sensitivity of the model outputs to each fit parameter by identifying parameter ranges that doubled the MSE of the fits. *Table 2* includes these parameter ranges. We also tested sensitivity to combinations of parameters using sloppy modeling (see further details below and *Brown and Sethna, 2003*). This approach identifies specific combinations of parameters that have maximal and minimal impact on model error – in our case the MSE. For cone models, parameters controlling the calcium feedback ($\beta$ and $K_{GC}$) could compensate for each other to have relatively minor effects on fit quality. Fit quality was particularly sensitive to changes in the parameters controlling the PDE activation ($\sigma$ and $\eta$). Rod models were relatively insensitive to changes in $\sigma$ and $\eta$ and were particularly sensitive to changes in $K_{GC}$ and $\gamma$.

## Sloppy modeling

Sloppy modeling enables us to understand how a model's cost function depends on the model parameters individually and in combination. We started by computing the Hessian matrix $H$ at a location in parameter space given by the consensus fit parameters

$$H_{i,j} = \frac{\partial C}{\partial \theta_i} \cdot \frac{\partial C}{\partial \theta_j}$$

Here, $\theta_i$ and $\theta_j$ represent different parameters and $C$ is the cost function. The Hessian provides a measure of the curvature of the cost function around the best-fit model parameters. We numerically approximate the Hessian by taking incremental changes in pairs of parameters around the best fit and measuring the changes in MSE. The eigenvectors of the Hessian matrix identify directions in parameter space in which the cost function changes slowly or rapidly.

## Linear model

The linear models used to generate the target response were obtained by fitting a parameterized linear filter to the response of the full model to low-contrast Gaussian noise at a specific mean light level. The linear filter is defined as:

$$L\left(t\right) = \alpha \left(\frac{t}{\tau_D}\right)^3 \cdot exp\left(-\frac{t}{\tau_R}\right) / \left(1 + \left(\frac{t}{\tau_D}\right)^3\right) \tag{7}$$

where $\alpha$ is a scaling factor, and $\tau_R$ and $\tau_D$ are the rising and decay time constants. This model provides good fits to measured low-contrast responses of rods and cones (*Angueyra and Rieke, 2013*). Photoreceptor responses to low-contrast stimuli depend linearly on the stimulus (*Hass et al., 2015*), and hence this approach ensured that the linear model matched the full model for such stimuli. The linear filter was convolved directly with the light stimulus to obtain a linear estimate of the responses.

## Inverse model

The differential equations that comprise our model consist of several time-independent nonlinearities (*Equations 4 and 6* in *Figure 1*), several linear differential equations (*Equations 1, 2, and 5* in *Figure 1*), and one nonlinear differential equation (*Equation 3* in *Figure 1*). The time-independent

nonlinearities can be inverted directly as a look-up table – for example the cGMP depends directly on the current as $G = (I/k)^{1/n}$. The linear differential equations can be solved directly by deconvolution – for example the Fourier transform of the calcium concentration from *Equation 5* in *Figure 1* is obtained directly from the Fourier transform of the measured current ($I(f)$) as

$$C(f) = \frac{qI(f)}{\beta - 2\pi if} \tag{8}$$

where $f$ is the temporal frequency. The nonlinear differential equation (*Equation 3* in *Figure 1*) can be reexpressed as an equation for the PDE activity $P$ in terms of the cGMP concentration $G$ and its derivative $dG/dt$ as

$$P(t) = \frac{S(t) - dG(t)/dt}{G(t)} \tag{9}$$

This equation can be solved given $G$ (obtained from the current via *Equation 4* in *Figure 1*), and $S$ (obtained from the calcium concentration $C(t)$ and *Equation 6* in *Figure 1*). These steps in the inversion process are illustrated in *Figure 3*.

This process results in an exact mapping between responses and stimuli – that is every response $I(t)$ has a corresponding unique stimulus $Stim(t)$. In practice, noise in the measured responses can corrupt the estimated stimuli. This is a general problem in deconvolution of noisy data – for example in microscopy. As indicated in the text, we controlled this noise when necessary by imposing that the power spectrum of our estimated stimulus be equal to the power spectrum of the true stimulus. This constraint was imposed by generating the estimated stimulus as described above, and then reweighting the power spectrum of the estimated stimulus to match the power spectrum of the true stimulus. This constraint was used in generating the estimated stimuli in *Figure 4*.

## Predictably altering photoreceptor responses

Model inversion as described above allowed us to identify stimuli that would produce a desired phototransduction current (which we refer to as a target response). For the applications in *Figures 6–10*, we generated this response using the linear model of the transduction cascade described above (*Equation 7*). We passed an initial stimulus (e.g. a sinusoid in *Figures 6 and 7*) through the linear model to generate the target response. The model inversion process for the full (i.e. nonlinear) phototransduction cascade model then determined the stimulus which, when delivered to the full model, would produce the linear target response. In particular, this allowed us to identify stimuli that negated the impact of adaptation on the photoreceptor responses.

## Acknowledgements

We thank Shellee Cunnington for excellent technical support. We are grateful for assistance from the WaNPRC staff, especially Chris English, for access to primate tissue. Shane Boniec, Greg Horwitz, and Saad Idrees provided invaluable feedback on a draft of the manuscript. This work was supported by NIH grant EY028542 (FR) and the Air Force Office of Scientific Research under award number FA9550-21-1-0230 (special thanks to the AFOSR's Cognitive & Computational Neuroscience Program).

## Additional information

### Competing interests

Fred Rieke: Reviewing editor, eLife. The other authors declare that no competing interests exist.

### Funding

| Funder | Grant reference number | Author |
| --- | --- | --- |
| National Institutes of Health | EY028542 | Fred Rieke |

| Funder | Grant reference number | Author |
|---|---|---|
| Air Force Office of Scientific Research | FA9550-21-1-0230 | Fred Rieke |

The funders had no role in study design, data collection, and interpretation, or the decision to submit the work for publication.

## Author contributions

Qiang Chen, Conceptualization, Data curation, Software, Formal analysis, Investigation, Visualization, Writing – review and editing; Norianne T Ingram, Investigation; Jacob Baudin, Raunak Sinha, Investigation, Writing – review and editing; Juan M Angueyra, Software, Formal analysis, Investigation, Writing – review and editing; Fred Rieke, Conceptualization, Software, Formal analysis, Funding acquisition, Investigation, Visualization, Writing – original draft, Project administration, Writing – review and editing

## Author ORCIDs

Juan M Angueyra ⓘ https://orcid.org/0000-0002-9217-3069
Raunak Sinha ⓘ https://orcid.org/0000-0002-7553-1274
Fred Rieke ⓘ https://orcid.org/0000-0002-1052-2609

## Ethics

Experiments were performed on mouse and primate retina following procedures in accordance with the University of Washington Institutional Animal Care and Use Committee (protocols 4140-01 and 3030-01). Primate retinas were obtained through the Tissue Distribution Program of the University of Washington's Regional Primate Research Center. Recordings were made from retinas from Macaca fascicularis, Macaca nemestrina, and Macaca mulatta of both sexes, aged 2–20 years. Recordings from mouse used animals of both sexes with ages of 6–12 weeks.

Reviewer #1 (Public Review): https://doi.org/10.7554/eLife.93795.3.sa1
Reviewer #2 (Public Review): https://doi.org/10.7554/eLife.93795.3.sa2
Reviewer #3 (Public Review): https://doi.org/10.7554/eLife.93795.3.sa3
Author response https://doi.org/10.7554/eLife.93795.3.sa4

# Additional files

## Supplementary files

• MDAR checklist

## Data availability

Data used in all the model fitting is available at https://doi.org/10.5061/dryad.q2bvq83vg. This includes all the source data for Figures 1–5 and associated figure supplements.

The following dataset was generated:

| Author(s) | Year | Dataset title | Dataset URL | Database and Identifier |
|---|---|---|---|---|
| Rieke F, Chen Q | 2024 | Predictably manipulating photoreceptor light responses to reveal their role in downstream visual responses | https://doi.org/10.5061/dryad.q2bvq83vg | Dryad Digital Repository, 10.5061/dryad.q2bvq83vg |

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
