## [Editor Report · eLife assessment]

This paper provides an **important** method that uses a computational model to predict photoreceptor currents in mammalian photoreceptors. By inverting the model, visual stimuli can be constructed to produce desired photoreceptor current responses. The authors provide **compelling** evidence that this approach can disentangle the effects of photoreceptor nonlinearities including light adaptation from downstream nonlinear processing, thus facilitating future studies of the higher visual system.

---

## [Referee Report · Reviewer #1 (Public Review)]

Summary:

This manuscript aims at a quantitative model of how visual stimuli, given as time-dependent light intensity signals, are transduced into electrical currents in photoreceptors of macaque and mouse retina. Based on prior knowledge on the fundamental biophysical steps of the transduction cascade and a relatively small number of free parameters, the resulting model is found to fairly accurately capture measured photoreceptor currents under a range of diverse visual stimuli and with parameters that are (mostly) identical for photoreceptors of the same type.

Furthermore, as the model is invertible, the authors show that it can be used to derive visual stimuli that result in a desired, predetermined photoreceptor response. As demonstrated with several examples, this can be used to probe how the dynamics of phototransduction affect downstream signals in retinal ganglion cells, for example, by manipulating the visual stimuli in such a way that photoreceptor signals are linear or have reduced or altered adaptation. This innovative approach had already previously been used by the same lab to probe the contribution of photoreceptor adaptation to differences between On and Off parasol cells (Yu et al, eLife 2022), but the present paper extends this by describing and testing the photoreceptor model more generally and in both macaque and mouse as well as for both rods and cones.

Strengths:

The presentation of the model is thorough and convincing, and the ability to capture responses to stimuli as different as white noise with varying mean intensity and flashes with a common set of model parameters across cells is impressive. Also, the suggested approach of applying the model to modify visual stimuli that effectively alter photoreceptor signal processing is thought-provoking and should be a powerful tool for future investigations of retinal circuit function. The examples of how this approach can be applied are convincing and corroborate, for example, previous findings that adaptation to ambient light in the primate retina, as measured by responses to light flashes, mostly originates in photoreceptors. Application of the approach by other labs is facilitated by the clear exposition and the listing of obtained optimal parameter values.

Weaknesses:

The model is impressive, but not perfect, including some small systematic differences between model predictions and measurements from held-out cells. The deviations likely (partly) reflect differences between cells used for parameter optimization and test cells, as stated in the text (though this is not fully proven), which has to be kept in mind when applying the model, in particular with the listed parameters.

---

## [Referee Report · Reviewer #2 (Public Review)]

Summary:

This manuscript proposes a modeling approach to capture nonlinear processes of photocurrents in mammalian (mouse, primate) rod and cone photoreceptors. The ultimate goal is to separate these nonlinearities at the level of photocurrent from subsequent nonlinear processing that occurs in retinal circuitry. The authors devised a strategy to generate stimuli that cancel the major nonlinearities in photocurrents. For example, modified stimuli would generate genuine sinusoidal modulation of the photocurrent, whereas a sinusoidal stimulus would not (i.e., because of asymmetries in the photocurrent to light vs. dark phases of a sinusoidal stimulus); and modified stimuli that could cancel the effects of light adaptation at the photocurrent level. Using these modified stimuli, one could record downstream neurons, knowing that any nonlinearities that emerge must happen beyond the stage of the photocurrent. This could be a useful method for separating nonlinear mechanisms across different stages of retinal processing and may be useful in vivo.

Strengths:

(1) This is a very quantitative and thoughtful approach and addresses a long-standing problem in the field: determining the location of nonlinearities within a complex circuit, including asymmetric responses to different polarities of contrast, adaptation, etc.

(2) The study presents data for two primary models of mammalian retina, mouse and primate, and shows that the basic strategy works in each case.

(3) Ideally, the present results would generalize to the work in other labs and possibly other sensory systems. The authors do provide evidence that a photocurrent model constructed from data in one set of cells can be used in a second set of cells.

Weaknesses:

(1) The model is limited to describing photoreceptor responses at the level of photocurrents, as opposed to the output of the cell, which takes into account voltage-dependent mechanisms, horizontal cell feedback, etc., as the authors acknowledge. It could be interesting to expand the model in the future to include factors that affect photoreceptor output beyond the stage of the photocurrent.

(2). It will be interesting to eventually test the impact of this work for in vivo experiments.

---

## [Referee Report · Reviewer #3 (Public Review)]

Summary:

The authors propose to invert a mechanistic model of phototransduction in mouse and rod photoreceptors to derive stimuli that compensate for nonlinearities in these cells. They fit the model to a large set of photoreceptor recordings, and show in additional data that the compensation works. This can allow to exclude photoreceptors as a source of nonlinear computation in the retina, as desired to pinpoint nonlinearties in retinal computation. The recordings made by the authors are impressive and I appreciate the simplicity and elegance of the idea. The data support the authors conclusions.

Strengths:

- The authors collected an impressive set of recordings from mouse and primate photoreceptors, which is very challenging to obtain.

- The other proposes to exploit mechanistic mathematical models of a well understood phototransduction to design light stimuli which compensate for nonlinearities.

- The authors demonstrate through additional experiments that their proposed approach works and is useful for offering insights into retinal computation.

- The biophysical modeling approach is well described.

---

## [Author Response]

The following is the authors’ response to the original reviews.

**Public Reviews:**

**Reviewer #1 (Public Review):**
Summary:This manuscript aims at a quantitative model of how visual stimuli, given as time-dependent light intensity signals, are transduced into electrical currents in photoreceptors of macaque and mouse retina. Based on prior knowledge of the fundamental biophysical steps of the transduction cascade and a relatively small number of free parameters, the resulting model is found to fairly accurately capture measured photoreceptor currents under a range of diverse visual stimuli and with parameters that are (mostly) identical for photoreceptors of the same type.Furthermore, as the model is invertible, the authors show that it can be used to derive visual stimuli that result in a desired, predetermined photoreceptor response. As demonstrated with several examples, this can be used to probe how the dynamics of phototransduction affect downstream signals in retinal ganglion cells, for example, by manipulating the visual stimuli in such a way that photoreceptor signals are linear or have reduced or altered adaptation. This innovative approach had already previously been used by the same lab to probe the contribution of photoreceptor adaptation to differences between On and Off parasol cells (Yu et al, eLife 2022), but the present paper extends this by describing and testing the photoreceptor model more generally and in both macaque and mouse as well as for both rods and cones.Strengths:The presentation of the model is thorough and convincing, and the ability to capture responses to stimuli as different as white noise with varying mean intensity and flashes with a common set of model parameters across cells is impressive. Also, the suggested approach of applying the model to modify visual stimuli that effectively alter photoreceptor signal processing is thought-provoking and should be a powerful tool for future investigations of retinal circuit function. The examples of how this approach can be applied are convincing and corroborate, for example, previous findings that adaptation to ambient light in the primate retina, as measured by responses to light flashes, mostly originates in photoreceptors.Weaknesses:In the current form of the presentation, it doesn't become fully clear how easily the approach is applicable at different mean light levels and where exactly the limits for the model inversion are at high frequency. Also, accessibility and applicability by others could be strengthened by including more details about how parameters are fixed and what consensus values are selected.

Thank you - indeed a central goal of writing this paper was to provide a tool that could be easily used by other laboratories. We have clarified and expanded four points in this regard: (1) we have stated more clearly that mean light levels are naturally part of inversion process, and hence the approach can be applied across a broad range of light levels (lines 292-297); (2) we have expanded our analysis of the high frequency limits to the inversion and added that expanded figure to the main text (new Fig 5); (3) we have included additional detail about our calibration procedures, including our calibration code, to facilitate transfer to other labs; and, (4) we have detailed the procedure for identification of consensus parameters (line 172-182, 191-199 and Methods section starting on line 831).

**Reviewer #2 (Public Review):**
Summary:This manuscript proposes a modeling approach to capture nonlinear processes of photocurrents in mammalian (mouse, primate) rod and cone photoreceptors. The ultimate goal is to separate these nonlinearities at the level of photocurrent from subsequent nonlinear processing that occurs in retinal circuitry. The authors devised a strategy to generate stimuli that cancel the major nonlinearities in photocurrents. For example, modified stimuli would generate genuine sinusoidal modulation of the photocurrent, whereas a sinusoidal stimulus would not (i.e., because of asymmetries in the photocurrent to light vs. dark changes); and modified stimuli that could cancel the effects of light adaptation at the photocurrent level. Using these modified stimuli, one could record downstream neurons, knowing that any nonlinearities that emerge must happen post-photocurrent. This could be a useful method for separating nonlinear mechanisms across different stages of retinal processing, although there are some apparent limitations to the overall strategy.Strengths:(1) This is a very quantitative and thoughtful approach and addresses a long-standing problem in the field: determining the location of nonlinearities within a complex circuit, including asymmetric responses to different polarities of contrast, adaptation, etc.(2) The study presents data for two primary models of mammalian retina, mouse, and primate, and shows that the basic strategy works in each case.(3) Ideally, the present results would generalize to the work in other labs and possibly other sensory systems. How easy would this be? Would one lab have to be able to record both receptor and post-receptor neurons? Would in vitro recordings be useful for interpreting in vivo studies? It would be useful to comment on how well the current strategy could be generalized.

We agree that generalization to work in other laboratories is important, and indeed that was a motivation for writing this as a methods paper. The key issue in such generalization is calibration. We have expanded our discussion of our calibration procedures and included that code as part of the github repository associated with the paper. Figure 10 (previously Figure 9) was added to illustrate generalization. We believe that the approach we introduce here should generalize to in vivo conditions. We have expanded the text on these issues in the Discussion (sections starting on line 689 and 757).

Weaknesses:(1) The model is limited to describing photoreceptor responses at the level of photocurrents, as opposed to the output of the cell, which takes into account voltage-dependent mechanisms, horizontal cell feedback, etc., as the authors acknowledge. How would one distinguish nonlinearities that emerge at the level of post-photocurrent processing within the photoreceptor as opposed to downstream mechanisms? It would seem as if one is back to the earlier approach, recording at multiple levels of the circuit (e.g., Dunn et al., 2006, 2007).

Indeed the current model is limited to a description of rod and cone photocurrents. Nonetheless, the transformation of light inputs to photocurrents can be strongly nonlinear, and such nonlinearities can be difficult to untangle from those occurring late in visual processing. Hence, we feel that the ability to capture and manipulate nonlinearities in the photocurrents is an important step. We have expanded Figure 10 to show an additional example of how manipulation of nonlinearities in phototransduction can give insight into downstream responses. We have also noted in text that an important next step would be to include inner segment mechanisms (section starting on line 661); doing so will require not only characterization of the current-to-voltage transformation, but also horizontal cell feedback and properties of the cone output synapse.

(2) It would have been nice to see additional confirmations of the approach beyond what is presented in Figure 9. This is limited by the sample (n = 1 horizontal cell) and the number of conditions (1). It would have been interesting to at least see the same test at a dimmer light level, where the major adaptation mechanisms are supposed to occur beyond the photoreceptors (Dunn et al., 2007).

We have added an additional experiment to this figure (now Figure 10) which we feel nicely exemplifies the approach. The approach that we introduce here really only makes sense at light levels where the photoreceptors are adapting; at lower light levels the photoreceptors respond near-linearly, so our “modified” and “original” stimuli as in Figure 10 (previously Figure 9) would be very similar (and post-phototransduction nonlinearities are naturally isolated at these light levels).

**Reviewer #3 (Public Review):**
Summary:The authors propose to invert a mechanistic model of phototransduction in mouse and rod photoreceptors to derive stimuli that compensate for nonlinearities in these cells. They fit the model to a large set of photoreceptor recordings and show in additional data that the compensation works. This can allow the exclusion of photoreceptors as a source of nonlinear computation in the retina, as desired to pinpoint nonlinearities in retinal computation. Overall, the recordings made by the authors are impressive and I appreciate the simplicity and elegance of the idea. The data support the authors' conclusions but the presentation can be improved.Strengths:- The authors collected an impressive set of recordings from mouse and primate photoreceptors, which is very challenging to obtain.- The authors propose to exploit mechanistic mathematical models of well-understood phototransduction to design light stimuli that compensate for nonlinearities.- The authors demonstrate through additional experiments that their proposed approach works.Weaknesses:- The authors use numerical optimization for fitting the parameters of the photoreceptor model to the data. Recently, the field of simulation-based inference has developed methods to do so, including quantification of the uncertainty of the resulting estimates. Since the authors state that two different procedures were used due to the different amounts of data collected from different cells, it may be worthwhile to rather test these methods, as implemented e.g. in the SBI toolbox (https://joss.theoj.org/papers/10.21105/joss.02505). This would also allow them to directly identify dependencies between parameters, and obtain associated uncertainty estimates. This would also make the discussion of how well constrained the parameters are by the data or how much they vary more principled because the SBI uncertainty estimates could be used.

Thank you - we have improved how we describe and report parameter values in several ways. First, the previous text erroneously stated that we used different fitting procedures for different cell types - but the real difference was in the amount of data and range of stimuli we had available between rods and cones. The fitting procedure itself was the same for all cell types. We have clarified this along with other details of the model fitting both in the main text (lines 121-130) and in the Methods (section starting on line 832). We also collected parameter values and estimates of allowed ranges in two tables. Finally, we used sloppy modeling to identify parameters that could covary with relatively small impact on model performance; we added a description of this analysis to the Methods (section starting on line 903).

- In several places, the authors refer the reader to look up specific values e.g. of parameters in the associated MATLAB code. I don't think this is appropriate, important values/findings/facts should be in the paper (lines 142, 114, 168). I would even find the precise values that the authors measure interesting, so I think the authors should show them in a figure/table. In general, I would like to see also the average variance explained by different models summarized in a table and precise mean/median values for all important quantities (like the response amplitude ratios in Figures 6/9).

We have added two tables with these parameters values and estimates of allowable ranges. We also added points to show the mean (and SD) across cells to the population figures and added those numerical values to the figure legends throughout.

- If the proposed model is supposed to model photoreceptor adaptation on a longer time scale, I fail to see why this can be an invertible model. Could the authors explain this better? I suspect that the model is mainly about nonlinearities as the authors also discuss in lines 360ff.

For the stimuli that we use we see little or no contribution of slow adaptation in phototransduction. We have expanded the description of this point in the text and referred to Angueyra et al (2022) which looks at this issue in more detail for primate cones (paragraph starting on line 280).

- The important Figures 6-8 are very hard to read, as it is not easy to see what the stimulus is, the modified stimulus, the response with and without modification, what the desired output looks like, and what is measured for part B. Reworking these figures would be highly recommended.

We have reworked all of the figures to make the traces clearer.

- If I understand Figure 6 correctly, part B is about quantifying the relative size of the response to the little first flash to the little second flash. While clearly, the response amplitude of the second flash is only 50% for the second flash compared to the first flash in primate rod and cones in the original condition, the modified stimulus seems to overcompensate and result in 130% response for the second flash. How do the authors explain this? A similar effect occurs in Figure 9, which the authors should also discuss.

Indeed, in those instances the modified stimulus does appear to overcompensate. We suspect this is due to differences in sensitivity of the specific cells probed for these experiments and those used in the model construction. We now describe this limitation in more detail (lines 524-526). A similar point comes up for those experiments in which we speed the photoreceptor responses (new FIgure 9B), and we similarly note that the cells used to test those manipulations differed systematically from those used to fit the model (lines 558-560).

**Recommendations for the authors:**

**Reviewer #1 (Recommendations For The Authors):**
I only have a few minor questions and suggestions for clarification.It hasn't become fully clear to me how general the model is when different mean light levels (on long-time scales) are considered. Are there slow adaptation processes not captured in the model that affect model performance? And how should one go about setting the mean light level when, for example, probing ganglion cells with a stimulus obtained through model inversion? Should it work to add an appropriate DC component to the current that is provided as input to the inverted model? (Presumably, deriving a stimulus and then just adding background illumination should not work, or could this be a good approximation, given a steady state that is adapted to the background?)

We have clarified in the main text that slow adaptation does not contribute substantially to responses to the range of stimuli we explored (lines 281-289). We have also clarified that the stimulus in the model inversion is specified in isomerizations per second - so the mean value of the stimulus is automatically included in the model inversion (lines 293-298).

Furthermore, a caveat for the model inversion seems to be the potential amplification of high-frequency noise. The suggested application of a cutoff temporal frequency seems appropriate, but data are shown only for a few example cells. Is this consistent across cells? (Given that performance between, e.g., mouse cones can vary considerably according to Fig. 4B?) I would also like to suggest moving the corresponding Supplemental Figure (4.1) into the main part of the manuscript, as it seems quite important.

We have added population analysis to the new Figure 5 (which was Figure 4 - Figure Supplement 1). We have also clarified that the amplification of high frequency noise is an issue only when we try to apply model inversion to measured stimuli. When we use model inversion to identify stimuli that elicit desired responses, the target responses are computed from a linear model that has no noise, so this is not a concern in applications like those in Figures 6-10.

Also, could the authors explain more clearly what the effect of the normalization of the estimated stimulus by the power of the true stimulus is? Does this simply reduce power at high frequency or also affect frequencies below the suggested cutoff (where the stimulus reconstruction should presumably be accurate even without normalization)?

Indeed this normalization reduces high frequency power and has little impact on low frequencies where the inversion is accurate; this is now noted in the text (line 363). As for amplification of high frequency noise (previous comment), the normalization by the stimulus power is only needed when inverting measured responses (i.e. responses with noise) and is omitted when we are identifying stimuli that elicit desired responses (e.g. in Figures 6-10).

While the overall performance of the model to predict photoreceptor currents is impressive, it seems that particular misses occur for flashes right after a step in background illumination and for the white-noise responses at low background illumination (e.g. Figure 1B). Is that systematic, and if so what might be missing in the model?

Indeed the model (at least with fixed parameters across stimuli) appears to systematically miss a few aspects of the photoreceptor responses. These include the latency of the response to a bright flash and the early flashes in the step + flash protocol in Figure 1B. Model errors for the variable mean noise stimulus (Figure 2) showed little dependence on time even when responses were sorted by mean light level and by previous mean level. Model errors did not show a clear systematic dependence on light level; this likely reflects, at least in part, the use of mean-square-error to identify model parameters. We have expanded our discussion of these systematic errors in the text (lines 164-166).

I was also wondering whether this is related to the fact that in Figure 9B, the gain in the modified condition is actually systematically higher when there is more background light. Do the authors think that this could be a real effect or rather an overcompensation from the model? (By the way, is it specified what "Delta-gain" really is, i.e., ratio or normalized difference?)

We suspect this is an issue with the sensitivity of the specific cells for which we did these experiments (i.e. variability in the gamma parameter between cells). This sensitivity varies between cells, and such variations are likely to place the strongest limitation on our ability to use this approach to manipulate responses in different retinas. We now note those issues in the Results (lines 523-526, 557-559 and 591-593) with reference to Figures 9 (previously Figure 8) and 10 (previously Figure 9), and describe this limitation more generally in the Discussion (section starting on line 649). We have also changed delta-gain to response ratio, which seemed more intuitive.

Maybe I missed this, but it seems that the parameter gamma is fitted in a cell-type-specific fashion (e.g. line 163), but then needs to be fixed for held-out cells. How was this done? Is there much variability of gamma between cells?

There is variability in gamma between cells, and this likely explains some of systematic differences between data and model (see above and Methods, lines 902-903). For the consensus models in Figure 2B, gamma was allowed to vary for each cell while the remaining consensus model parameters were fixed. Gamma was set equal to the mean value across cells for model inversion (i.e. for all of the analyses in Figures 4-10). We have described the fitting procedure in considerably more detail in the revised Methods (starting on line 832).

For completeness, it would be nice to have the applied consensus model parameters in the manuscript rather than just in the Matlab code (especially since the code has not been part of the submission). Also, some notes on how the numerical integration of the differential equations was done would be nice (time step size?).

We have added tables with consensus parameters and estimates of the sensitivity of model predictions to each parameter. We have also added additional details about the numerical approaches (including the time step) to Methods.

Similarly, it would be nice to explicitly see the relationships that are used to fix certain model parameters (lines 705ff). And can the constants k and n (lines 709-710) be assumed identical for different species and receptor types?

We have added more details to the model fitting to the methods, including the use of steady-state conditions to hold certain parameters fixed (lines 862 and 866). We are not aware of any direct comparisons of k and n across species and receptor types. We have noted that model performance was not improved by modest changes in these parameters (due to compensation by other model parameters). More generally, we have explained how some parameters trade for others and hence the logic of fixing some even when exact values were not available.

For the previous measurements of m and beta (lines 712-713), is there a reference or source?

We have added references for these values.

Did the authors check for differences in the model parameters between cone types (e.g., S vs. M)?

We did not include S cones here. They are harder to record from and collecting a fairly large data set across a range of stimuli would be challenging. Our previous work shows that S cones have slower responses than L and M cones, and this would certainly be reflected in differences in model parameters. We have noted this in the text (Methods, line 808-810).

For the stated flash responses time-to-peak (lines 183-184), is this for a particular light intensity with no background illumination?

Those are flashes from darkness - now noted in the text.

Figure 2 - Supplement 1 doesn't have panel labels A and B, unlike the legend.

Fixed - thank you.

**Reviewer #2 (Recommendations For The Authors):**
(1) Fig. 2B - for some cells, the consensus model seems to fit better than the individual model. How is this possible?

This was mostly an error on our part (we inadvertently included responses to more stimuli in fitting the individual models, which slightly hampered their performance). Even with this correction, however, a few cells remain for which the consensus model outperforms and individual model. We believe this is because there is more data to constrain model parameters for the consensus models (since they are fit to all cells at the same time), and that can compensate for improvements associated with customizing parameters to specific cells.

(2) Fig. 2 Supplement 1, it would be useful to see a blow-up of the data in an inset, as in Fig. 2B.

Thanks - added.

(3) Line 400 - this paragraph could include additional quantification and statistics to back up claims re 'substantially reduced', 'considerably lower'.

We quantify that in the next sentence by computing the mean-square-error between responses and sinusoidal fits (also in Figure 7B, which now includes statistics as well). We have made that connection more direct in the text.

(4) Maybe a supplement to Fig. 8 could show the changes to the stimulus required to alter the kinetics in both directions - to give more insight into part B., especially.

Good suggestion - we have added the stimuli to all of the panels of the figure (now Figure 9).

(5) Fig. 8B - in 'Speed response up' condition - there seems to be error in the model for the decay time of the response - especially for the 'original' condition, which is not quantified in 8C. Was it generally difficult to predict responses to flashes?

That seems largely to reflect that the cells used for those experiments had faster initial kinetics than the average cells (responses to the control traces are also faster than model predictions in these cells - black traces in Figure 9B). We have added this to the text.

(6) Line 678, possibly notes that 405 nm equally activates S and M photopigments in mice, since most of the cones co-express the two photopigments (Rohlich et al., 1994; Applebury et al., 2000; Wang et al., 2011).

Thanks - we have added this (lines 827-829).

(7) The discussion could include a broader description of the various approaches to identifying nonlinearities within retinal circuitry, which include (incomplete list): recording at multiple levels of the circuit (e.g., Kim and Rieke 2001; Rieke, 2001; Baccus and Meister, 2002; Dunn et al., 2006; 2007; Beaudoin et al., 2007; Baccus et al., 2008); recording currents vs. spiking responses in a ganglion cell (e.g., Kim and Rieke, 2001; Zaghloul et al., 2005; Cui et al., 2016); neural network modeling approaches (e.g., Maheswaranathan et al., 2023); optogenetic approaches to studying filtering/nonlinear behavior at synapses (e.g., Pottackal et al., 2020; 2021).

Good suggestion - we have added this to the final paragraph of the Discussion.

**Reviewer #3 (Recommendations For The Authors):**
- I am personally not a fan of the style: "... as Figure 4A shows..." or comparable and much prefer a direct "We observe that X is the case (Figure 4A)". If the authors agree, they may want to revise their paper in this way.

We have revised the text to avoid the “... as Figure xx shows” construction. We have retained multiple instances which follow a “Figure xx shows that …” construction (which is both active rather than passive and does not use a personal pronoun).

- I am not a fan of the title. Light-adaption clamp caters only to a very specialized audience.

We have changed the title to “Predictably manipulating photoreceptor light responses to reveal their role in downstream visual responses.”

- The parameter fitting procedure should not only be described in Matlab code, but in the paper.

Thanks - we have expanded this in the Methods considerably (section starting on line 832).

- The authors should elaborate on why different fitting procedures were used.

We did not describe that issue clearly. The fitting procedures used across cells were identical, but we had different data available for different cell types due to experimental limitations. We have substantially revised that part of the main text to clarify this issue (paragraph starting on line 121).

- The authors state in line 126 that the input stimulus is supposed to mimic eye movements mouse, monkey, or human? Please clarify.

Thanks - we have changed this sentence to “abrupt and frequent changes in intensity that characterize natural vision.”

- Please improve the figure style. For example, labels should be in consistent capitalization and ideally use complete words (e.g. Figure 2B, 4B, and others).

We have made numerous small changes in the figures to make them more consistent.

- Is the fraction of variance calculated on held-out-data? Linear models should be added to Figure 2B.

The fraction of variance explained was not calculated on held out data because of limitations in the duration of our recordings. Given the small number of free parameters, and the ability of the model to capture held out cells, we believe that the model generalizes well. We have added a supplemental figure with linear model performance (Figure 2 - Figure Supplement 2).

- Fig. 9A is lacking bipolar cell and amacrine cell labels. Currently, it looks like HC is next to the BC in the schematic.

Thanks - we have updated that figure (now Figure 10A)

- Maybe I am misunderstanding something, but it seems like the linear model prediction shown in Figure 2A for the rod could be easily improved by scaling it appropriately. Is this impression correct or why not?

We have clarified how the linear model is constructed (by fitting the linear model to low contrast responses of the full model at the mean stimulus intensity). We also added a supplemental figure, following the suggestion above, showing the linear model performance when a free scaling factor is included for each cell.

- The verification experiment in Fig. 5 is only anecdotal and is elaborated only in Figure 6. If I am not mistaken, this does not necessitate its own figure/section but could rather be merged.

We have kept this figure separate (now Figure 6) as we felt that it was important to highlight the approach in general in a figure before getting into quantification of how well it works.

- Figure 5 right is lacking labels. What is red and grey?

Thanks for catching that - labels are added now.

- The end of the Discussion is slightly unusual. Did some text go missing?

Thanks - we have rearranged the Discussion so as not to end on Limitations.

- There is a bonus figure at the end which seems also not to belong in the manuscript.

Thanks - the bonus figure is removed now.

- The methods should also describe briefly what kind of routines were used in the Matlab code, e.g. gradient descent with what optimizer?

We’ve added that information as well.